# Discrepancies between observations and climate models of large-scale wind-driven Greenland melt influence sea-level rise projections

Dániel Topál [1,2,3] ✉, Qinghua Ding [2] ✉, Thomas J. Ballinger [4] ✉, Edward Hanna[5], Xavier Fettweis [6], Zhe Li[2] & Ildikó Pieczka[7]

While climate models project that Greenland ice sheet (GrIS) melt will continue to accelerate with climate change, models exhibit limitations in capturing observed connections between GrIS melt and changes in high-latitude atmospheric circulation. Here we impose observed Arctic winds in a fully-coupled climate model with fixed anthropogenic forcing to quantify the influence of the rotational component of large-scale atmospheric circulation variability over the Arctic on the temperature field and the surface mass/energy balances through adiabatic processes. We show that recent changes involving mid-to-upper-tropospheric anticyclonic wind anomalies – linked with tropical forcing – explain half of the observed Greenland surface warming and ice loss acceleration since 1990, suggesting a pathway for large-scale winds to potentially enhance sea-level rise by ~0.2 mm/year per decade. We further reveal fingerprints of this observed teleconnection in paleo-reanalyses spanning the past 400 years, which heightens concern about model limitations to capture wind-driven adiabatic processes associated with GrIS melt.

Greenland ice sheet (GrIS) melt, a significant source of current barystatic sea-level rise, has accelerated over recent decades and constitutes a key indicator of human influence on the climate system[1–5]. Indeed, the anthropogenically forced response—multi-model means, or single-model large ensemble means—of GrIS surface conditions in CMIP5/6 climate models is mostly consistent with GrIS surface temperature and mass balance changes derived from satellite-based observations and reanalyses[2,6–8]. However, it seems that observed melting processes are not fully captured in CMIP5/6 and that some of the models produce a portion of GrIS warming for wrong reasons. Previous studies suggest that the observed increase in the frequency and persistence of anticyclonic circulation (blocking) anomalies since about 1990 has contributed to enhanced GrIS melt over the last three decades[9–14]. On the contrary, climate models (both single-model ensembles[15] and individual models[16–19]) fail to depict changes in blocking conditions associated with surface temperature warming around much of the GrIS, which raises concerns about existing discrepancies between the mechanisms causing GrIS melt in the observational and model worlds. These uncertainties hinder the quantification of the GrIS's sensitivity to anthropogenic forcing and affect sea-level rise projections[4,7].

[1]Institute for Geological and Geochemical Research, Research Centre for Astronomy and Earth Sciences, MTA-Centre for Excellence, ELKH, Budapest, Hungary. [2]Department of Geography and Earth Research Institute, University of California-Santa Barbara, Santa Barbara, CA, USA. [3]ELTE Eötvös Loránd University, Doctoral School of Environmental Sciences, Budapest, Hungary. [4]International Arctic Research Center, University of Alaska Fairbanks, Fairbanks, AK, USA. [5]Department of Geography and Lincoln Climate Research Group, University of Lincoln, Lincoln, UK. [6]SPHERES research units, Geography Department, University of Liège, Liège, Belgium. [7]ELTE Eötvös Loránd University, Institute of Geography and Earth Sciences, Department of Meteorology, Budapest, Hungary. ✉e-mail: topal.daniel@csfk.org; qinghua@ucsb.edu; tjballinger@alaska.edu

Anthropogenic emissions driven diabatic warming—i.e. radiative forcing, which is further amplified by the surface-albedo feedbacks in the ablation zone and widespread cloud radiative effects[20–24]—and processes related to large-scale atmospheric and oceanic circulation variability together shape the recent changes of the GrIS[9–13,17,25–28]. Regarding the physical mechanisms, circulation-driven surface friction is key to creating adiabatic warming around the core of the anticyclone, which together with advective effects cause enhancement of longwave radiation anomalies (both clear-sky and cloud radiative) in the lower troposphere[21,22,29–31]. Synchronously, beneath the centre of the high-pressure, clear skies favour incoming shortwave radiation anomalies that tend to amplify GrIS surface melt especially in coastal regions with their lower albedo[12,20,24]. The regional circulation-driven process is further suggested to be sensitive to tropical forcing through Rossby-wave activity excited by anomalous sea surface temperatures (SST) in the tropical Pacific[32–35].

Current understanding suggests that GrIS surface conditions are determined by a quantitatively yet uncertain combination of (i) atmospheric circulation changes primarily via the aforementioned adiabatic warming process and (ii) anthropogenic diabatic warming, which may also cause wind changes due to proportional relationships between air temperature and pressure. Nonetheless, previous analyses of state-of-the-art climate model simulations indicate weak Arctic wind changes in response to anthropogenic forcing and suggest forced temperature changes are vertically rather uniform in the summer season[15,27]. Although one cannot rule out that model biases impact the forced temperature response in the Arctic, the observed mid-to-upper-level wind-driven process, which causes vertically non-uniform temperature changes[15,27], is an acknowledged contributor in shaping GrIS surface conditions during recent decades[9–14].

Given the above considerations, we hypothesize that the separation of the physical mechanisms behind GrIS surface warming into adiabatic (causing vertically non-uniform warming) and diabatic (causing vertically uniform warming) processes offers the potential for not only showcasing the divergent physical mechanisms behind observed and modelled GrIS warming, but also for quantifying the contribution of large-scale winds to accelerating sea-level rise. This is only possible by taking a conceptually different, dynamical approach contrary to previous studies that have mainly applied diagnostic and statistical approaches to observations and model simulations forced by constant or varying greenhouse gases over the past 40–150 years when examining the effects of atmospheric circulation on GrIS melt[11–13,25,26].

Here we impose observed winds from the surface to the top-of-the-atmosphere into a fully coupled model (CESM1), while keeping anthropogenic forcing-fixed, and compare our 10-member wind-nudging experiment to the CESM1 40-member large ensemble (CESM-LE[36]). This comparison reveals the importance of observed winds in causing the recent acceleration of GrIS melt and corresponding increase in the rate of barystatic sea-level rise relative to simulated GrIS changes in the ensemble mean (forced response) and the ensemble spread (internal variability) in the CESM-LE. While nudged winds may show an anthropogenic forcing signature, the comparison between our nudging-run and the CESM-LE mean may more clearly reveal the relative importance of internal/external forcing of observed wind changes. Further, forced wind changes might be affected by model structural biases, which complicates the quantification of the internal/forced observed wind changes. Therefore, we compare our nudging experiment to simulations of 31 climate models participating in the Coupled Model Inter-comparison Phase 5 (CMIP5), which helps to clarify whether any structural bias in the CESM-LE may be common across other models. In addition, we extend our analysis beyond the observational era by utilizing two recently available paleoclimatic proxy-data-assimilated reconstructions (paleo-reanalyses) in addition to independent Greenland ice core and oceanic coral proxy records spanning the past 400 years. These analyses offer the

opportunity to derive further insights into the persistence and robustness of the observed wind-driven GrIS warming mechanisms during periods with much less influence from anthropogenic emissions. By doing so we expect to advance the current understanding of large-scale atmospheric forcing driven GrIS surface changes since the early 17th century and contribute to contextualizing uncertainties of current model projections of sea-level rise.

## Results

### Observed and modelled summer GrIS melt and overlying circulation changes

We first describe past changes in GrIS surface conditions using mass balance estimates from the Ice Sheet Mass Balance Inter-comparison Exercise (IMBIE[6]) in addition to surface mass balance (SMB, Eq. 1 in Methods) and surface air temperature (SAT) simulations from a widely used and GrIS optimized[37–39] regional climate model, Modéle Atmosphérique Régional, (MAR, Methods), which is 6-hourly forced by the ERA5[39,40] reanalysis at its boundaries. We also characterize synchronous changes in the overlying atmospheric circulation in ERA5 since 1980 alongside increasing GrIS mass loss[3,6].

Both the satellite-observed rate of annual total mass change[6] and MAR climate model simulations of summer (June–July-August, JJA) SAT and SMB anomalies over Greenland show substantial interannual variability during 1980–2018 (Fig. 1a). Similar year-to-year variability is observed with sea surface temperatures (SST, Methods) over Baffin Bay (60°–80°N; 50°–70°W) ($r_{SST;SAT} = 0.78$ (0.67); $r_{SST;SMB} = 0.85$ (0.77) for raw (linearly detrended) data), indicating a possible shared driving mechanism behind Greenland and proximate ocean surface changes beyond anthropogenic forcing (Fig. 1b). We further reveal concomitant changes in atmospheric circulation over the GrIS by re-creating an often used metric—the Greenland Blocking Index (GBI[14,16,41], Methods) —and developing the Greenland Streamfunction Index (GSI, Fig. 1c). The GSI is calculated from the ERA5 500 hPa streamfunction ($\Psi500$) by spatially averaging the area-weighted $\Psi500$ anomaly field over the GrIS (20°–80°W; 60°–80°N) in JJA (Methods). The GSI is strongly correlated (r-0.88) with the GBI in summer, but because it reflects the rotational features of large-scale circulation that control the adiabatic warming process in the lower troposphere, it better represents the physical mechanisms discussed herein (Methods).

The ERA5 GSI explains significant part of the interannual variability in MAR-simulated summer SAT/SMB and Baffin Bay SSTs over 1980–2018, regardless of whether the underlying trend is removed or maintained, suggesting that this relationship results primarily from internal variability ($r^2_{GSI;SAT} = 0.56$ (0.46); $r^2_{GSI;SMB} = 0.69$ (0.61); $r^2_{GSI;SST} = 0.56$ (0.53) for raw (linearly detrended) data). The accompanying spatial trends in the MAR-simulated GrIS SAT (0.446 K decade$^{-1}$) and circulation—as described by ERA5 500hPa geopotential height (Z500; 12–15 m decade$^{-1}$) and horizontal winds (GSI; 0.844 10$^6$ m$^2$ s$^{-1}$ decade$^{-1}$)—show synchronous changes over 1980–2018. Such trends suggest that the aforementioned high-pressure-driven adiabatic warming process has likely acted in concert with anthropogenic forcing in shaping GrIS climate variability over the past four decades (Fig. 1d).

Contrary to the observed synchrony between GrIS surface and overlying atmospheric circulation changes, 31 CMIP5 climate models and the 40-member CESM1-LE also indicate less of an influence from wind changes on GrIS warming (Supplementary Fig. 1). It is also the case in version 2 of CESM[42] and more generally in the CMIP6 models[4,11]. These findings suggest the possibility that climate models may misrepresent driving mechanisms; hence, to further interpret the potential consequences, we conduct a wind-nudging model experiment (see Methods).

### Separating diabatic vs. adiabatic mechanisms driving GrIS melt

To distinguish between the two dominant mechanisms of observed GrIS summer warming as manifested in the circulation-driven

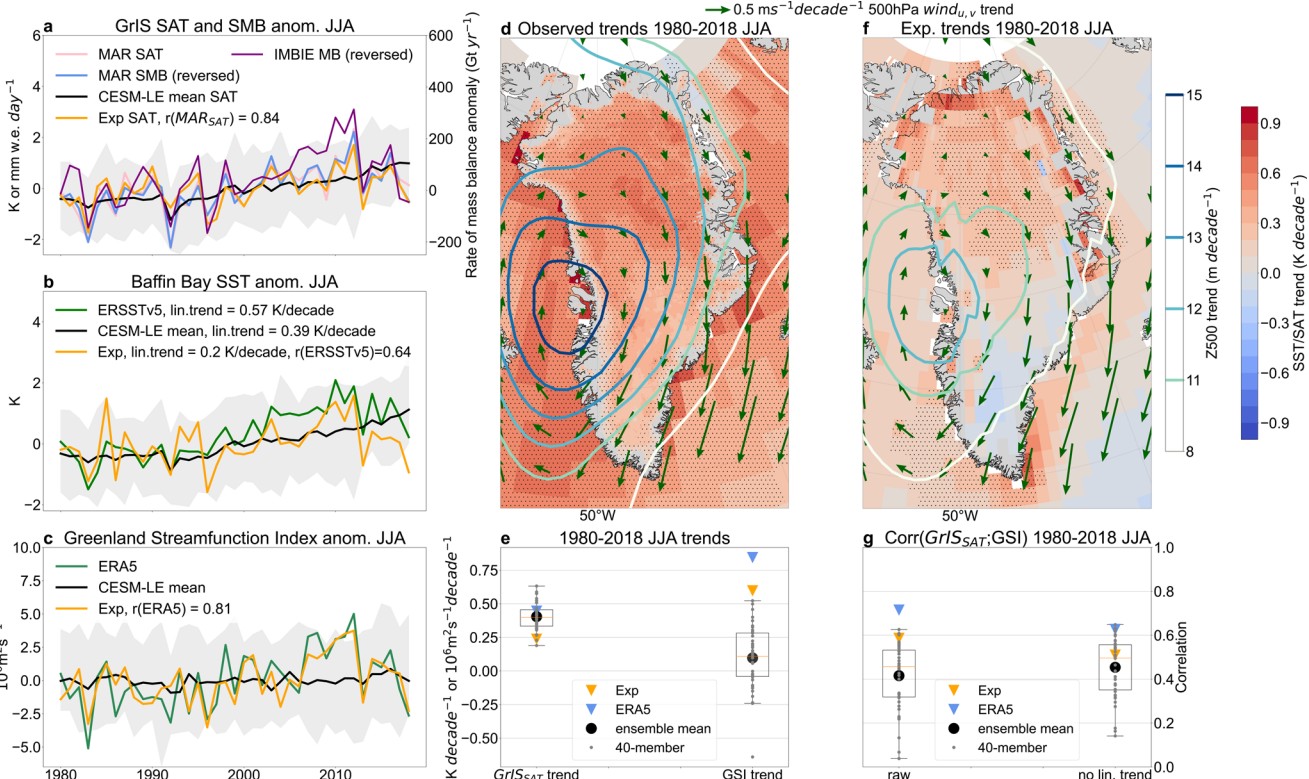

**Fig. 1 | Observed and simulated circulation-driven GrIS summer surface conditions over 1980–2018. a** Greenland ice sheet (GrIS) spatial-averaged time series of anomalies in MAR surface air temperature (SAT, pink) and surface mass balance (SMB, blue), the ensemble mean SAT from CESM1 large ensemble (CESM-LE, black) and the SAT from the wind-nudging experiment (Exp) in CESM1 (orange) in summer (June–July–August, JJA) as well as the observed annual rate of total GrIS mass change (IMBIE) (purple). The grey shading in **a** represents the range of all members' SAT anomalies from the CESM-LE. In **b**, ERSSTv5 sea surface temperatures (SST) averaged over the Baffin Bay (60°–80°N; 50°–70°W) (green) is compared with the ones calculated using the CESM-LE mean (with black), the spread in CESM-LE (grey shading) and the wind-nudging experiment (Exp, orange) for JJA. **c** is same as **b**, but for the 500hPa Greenland streamfunction index (GSI, Methods). In **d**, spatial maps of the linear trend in MAR SAT (shading), ERSSTv5 SSTs (shading), 500 hPa geopotential height (Z500, contours) and 500 hPa horizontal winds (arrows) from ERA5 in JJA for the 1980–2018 period are shown. Panel **f** is the same as **d** but from the wind-nudging experiment with the CESM1 (Exp, Methods). Panel **e** shows the linear trends in GrIS spatial-averaged SAT (first box-and-whiskers plot), the GSI (second box-and-whiskers plot) and **g** the raw (detrended) correlations between GrIS SAT and GSI in the first (second) box-and-whiskers plots as shown with markers corresponding to the legend. The whiskers extend to 1.5 times the interquartile-range (IQR) and the median is indicated with an orange horizontal line, outliers (that extend 1.5 IQR) with crosses. Hatching in **d** and **f** indicate areas with statistically significant linear trends based on the Mann–Kendall test ($p < 0.05$).

adiabatic component (vertically non-uniform warming) and the radiative forcing-induced diabatic warming (vertically uniform warming), we take a two-step approach in a dynamical modelling framework: first, we use the fully coupled Community Earth System Model 1.2 (nominal 1 degree resolution), and then we employ the Community Ice Sheet Model (CISM) Glimmer with a higher spatial resolution (~5 km) to conduct atmospheric wind-nudging experiments without interaction from time-varying anthropogenic forcing. To do so, we set external forcing (greenhouse gases, aerosols, solar) to constant values at the level of the year 2000 (367 ppm), which roughly represent the climatological mean values over 1980–2018. This allows us to directly compare the nudging experiment with ERA5, MAR and the CESM-LE to reveal the relative responses of the GrIS to atmospheric circulation changes and greenhouse gas forcing (see further details in Methods).

Constraining winds in the CESM1 results in the model closely resembling the interannual variability in ERA5 GSI (Fig. 1c, r = 0.81), in MAR GrIS SAT (Fig. 1a, r = 0.84) and in ERSSTv5 Baffin Bay SSTs (Fig. 1b, r = 0.64). As for the summertime spatial trend patterns, on average, 53% of observed SAT, 74% of the Z500 and 35% of the Baffin Bay SST changes between 1980–2018 are captured in the nudging experiment (Fig. 1d, f). A further comparison between our nudging-run-derived GrIS surface conditions and those simulated by each of the CESM-LE members reinforces the differences between the simulated and

observed sensitivity of the ice sheet to wind changes seen in other CMIP models[4,11,42] (Supplementary Fig. 1). We highlight the contrast between the summer GrIS SAT/GSI trends over 1980–2018 in the observations and simulated in individual members comprising the CESM-LE, which underestimate the ERA5 GSI trend while encompassing the observed SAT trend during 1980–2018 (Fig. 1e). Furthermore, the correlations between GSI and GrIS spatial-averaged SAT are weaker in each individual CESM-LE member than in ERA5 (r~0.75), and only three of these members exhibit equal or greater correlations compared to the nudging experiment (r~0.6) (Fig. 1g).

To account for observed wind changes that may stem from anthropogenic forcing we examine wind changes in the CESM-LE mean and in the multi-model mean of 31 CMIP5 models. Although the CESM-LE (CMIP5) mean GrIS SAT trend is ~90% (~60%) of its ERA5 counterpart, the equivalent value for the GSI is only ~10% (~2%). Compared to the significant wind changes in ERA5, the year-to-year GSI series (Fig. 1c, Supplementary Fig. 1b) show only subtle changes over 1980–2018 in both CESM-LE and CMIP5 ensemble means. These findings are similar to both the weak trend in 500 hPa horizontal winds in the CESM-LE mean (Supplementary Fig. 2a) and the uniform geopotential height response simulated by other single-model large ensembles[15]. While <10% of observed wind changes may be affected by anthropogenic forcing, a note of caution is warranted as models may also suffer from structural biases that can obstruct a

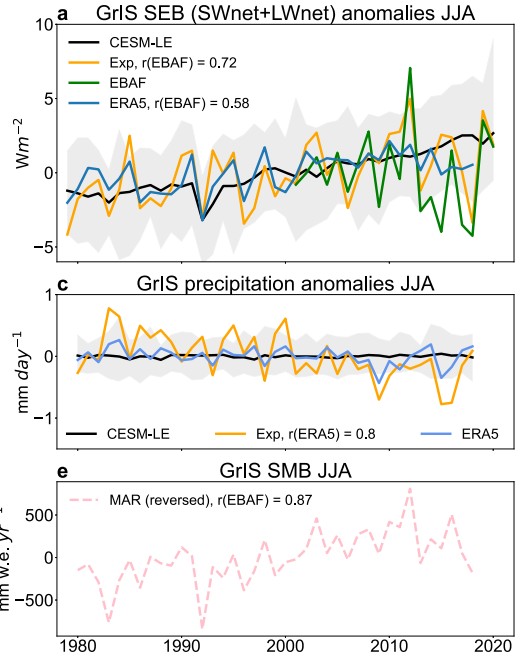
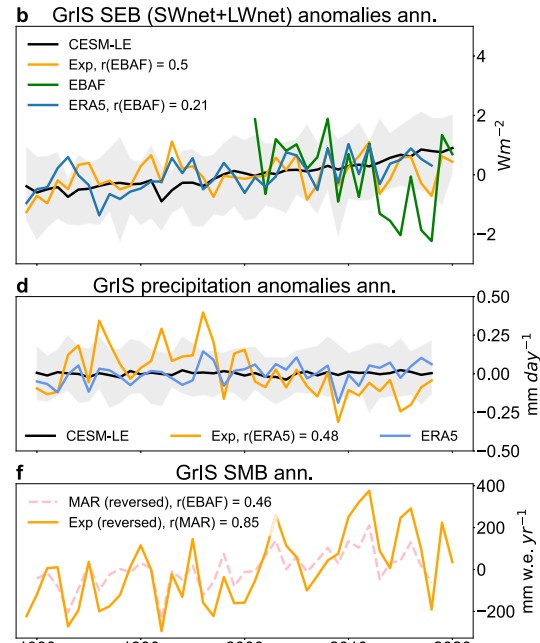

**Fig. 2 | The critical role of circulation in determining observed GrIS surface mass/energy balance over 1980–2018 as simulated by the nudging experiment.** Comparisons between the Greenland ice sheet (GrIS) spatial-averaged surface energy balance (SEB) anomaly time series in ERA5 (blue), EBAF satellite product (green, Methods), the nudging experiment (Exp, orange) and the CESM-LE (grey shading and black line for the ensemble mean) for **a** summer (June–July–August, JJA) and **b** for annual means. Comparison between ERA5 (blue), the nudging experiment (Exp, orange) and the CESM-LE (grey shading and black line for the ensemble mean) GrIS spatial-averaged precipitation for **c** JJA and **d** the annual means. Panel **e** shows the MAR-simulated GrIS surface mass balance (SMB) anomalies for JJA along with **f** showing the comparison of annual SMB between MAR and the nudging experiment involving CISM-Glimmer. In the legends, 'r()' indicates the correlation coefficients between the corresponding variable and the one in the bracket.

realistic wind response to anthropogenic emissions, and this shortcoming may limit the certainty of calculations that are currently available.

We further interpret the implications of the lack of circulation changes in models for GrIS melt-driven processes and assess SMB changes, which are directly linked to sea-level rise. First, we separately assess the surface energy balance (SEB; Eq. 2 in Methods) and total precipitation in the nudging simulation, which are the two key SMB components[43] (Eq. 1 in Methods). Similar to what we have seen for the GrIS SAT, imposing observed winds in our model experiment is sufficient to simulate a close match with both the summer and annual SEB interannual variabilities (Fig. 2a, b) seen in the ERA5 (1980–2018) and the satellite product (CERES-EBAF[44], 2001–2020; Methods). As for interdecadal time scales, the linear trends in the SEB also indicate an overall adequate simulation of satellite-observed changes in the nudging experiment in JJA (Supplementary Fig. 3a, b) and annual means (Supplementary Fig. 3d, e). These results suggest a decisive role for atmospheric circulation in controlling GrIS climate variability from an energy balance perspective. However, in line with the above-demonstrated limitations of the CESM-LE in capturing the circulation-related adiabatic component of GrIS melt, all members of the CESM-LE show relatively small year-to-year coupling between SEB and the GSI compared with ERA5 and the nudging simulation (Supplementary Fig. 4b) and show SEB changes with opposite sign compared with the EBAF over 2001–2020 (Supplementary Fig. 3c, f, g). As for the total precipitation averaged over the GrIS, the nudging experiment performs reasonably well in replicating ERA5 summer and annual mean precipitation (Fig. 2c, d). Although our experiment seems to overestimate precipitation variability over the GrIS, especially before 2000 (-2.8 (-1.6) times greater standard deviation between 1980–2000 (2000–2018)), we conclude that precipitation is highly sensitive to imposed winds in the nudging-run and, therefore, in reality.

To shed light on the contribution of wind-driven circulation directly to SMB variability, we force Glimmer-CISM with the daily output from one member of the nudging experiment (Methods). The resulting Glimmer-CISM annual mean GrIS SMB exhibit similar climatology to the ERA5-forced MAR simulation, albeit with overestimating the ablation zone melt during 1980–2018 (Supplementary Fig. 6). Despite the climatological bias, our experiment qualitatively reproduces the observed features of SMB variability of the past four decades, as shown by the comparison between the spatially averaged SMB anomaly time series from our Glimmer-CISM experiment and the ERA5-forced MAR simulation (Fig. 2f; r = 0.85; Supplementary Fig. 5c).

## Wind-driven GrIS mass loss and sea-level rise acceleration
We have seen that, alongside anthropogenically forced diabatic warming, atmospheric circulation changes are an important influence on the energy budget, precipitation and hence surface mass balance of the ice sheet. Besides the total cumulative ice loss from the GrIS, that has led to $10.8 \pm 0.9$ mm increase in global mean sea-levels over the past three decades[6], an additional key aspect is the observed acceleration in the rate of GrIS mass loss ($-132.8$ Gt yr$^{-1}$ decade$^{-1}$) over 1990–2012[6] that was a dominant contributor to enhanced sea-level rise[45,46] ($1.2 \pm 0.7$ mm yr$^{-1}$ decade$^{-1}$ over 1993–2017). During 1990–2012, both the ERA5 and the nudging-derived GSI show concomitant increase, followed by a levelling-off until 2018 (Fig. 1c), which corresponds with the lack of tropical forcing and a generally positive North Atlantic Oscillation phase (and less Greenland Blocking) linked to colder conditions over Greenland[12,34]. Quantifying the contribution of wind-driven circulation to increased GrIS melt and resulting sea-level rise over 1990–2012 is crucial to better contextualize not only the misrepresentation of the adiabatic/diabatic driving mechanisms in the CESM-LE, but also identify the future potential of atmospheric circulation to further amplify as well as to counteract anthropogenically forced GrIS changes.

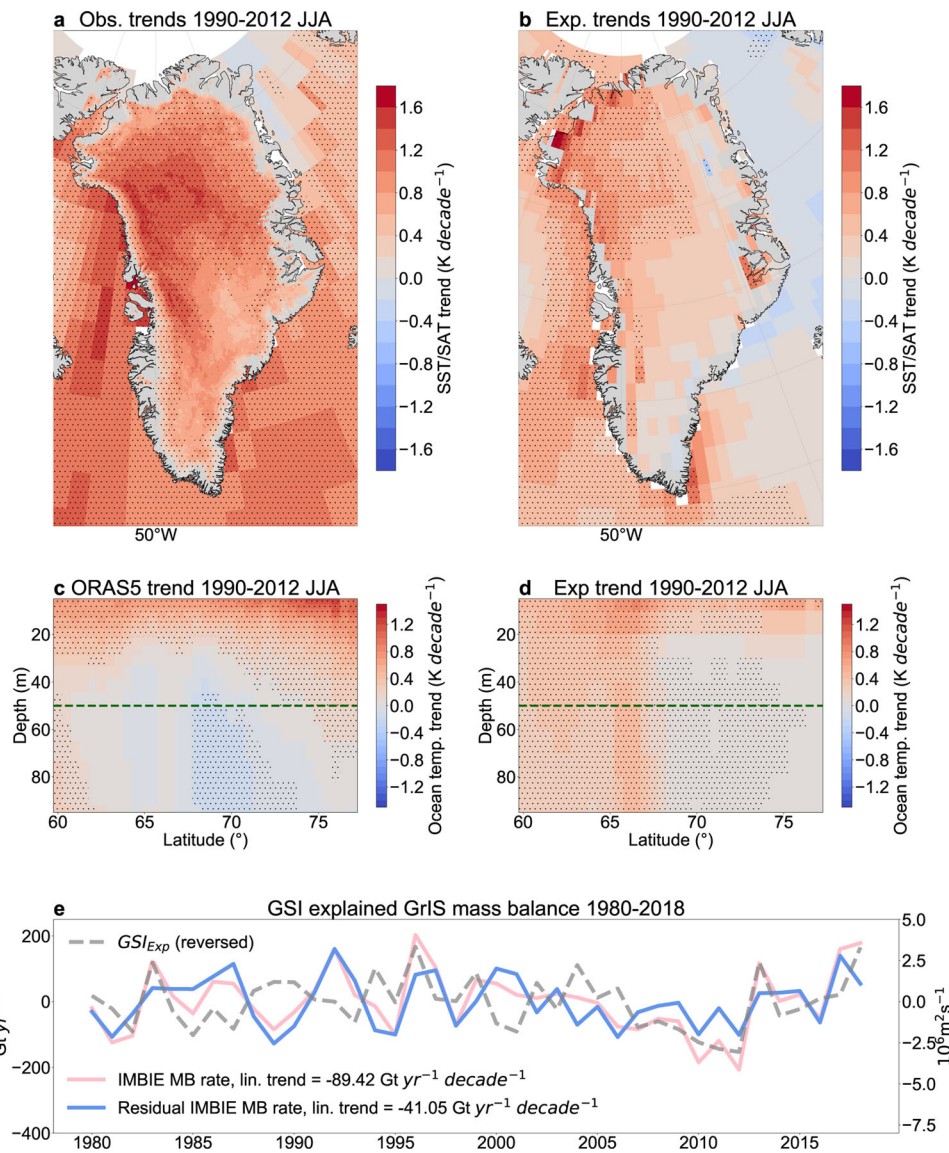

**Fig. 3 | The wind-driven acceleration of Greenland ice loss and Baffin Bay upper-ocean warming between 1990 and 2012.** Linear trends in summer (June–July–August, JJA) **a** ERA5 surface air temperature (SAT) and ERSSTv5 sea surface temperature (SST), **b** the nudging experiment-derived (Exp) SAT/SST, **c** ORAS5 upper-ocean (0–100 m) temperature (zonal mean over 50°W–70°W longitudinal band) and **d** nudging experiment-derived upper-ocean (0–100 m) temperature (zonal mean over 50°W–70°W longitudinal band) between 1990 and 2012. The green dashed lines in **c**, **d** indicate the section used to average ocean temperatures shown in Supplementary Fig. 7b. Hatching indicates statistically significant ($p < 0.05$) trends. Panel **e** shows the linearly detrended annual mean rate of Greenland mass balance (MB) anomaly from The IMBIE Team[6] (pink) and the linearly detrended summer 500 hPa Greenland streamfunction index derived from the nudging experiment ($GSI_{Exp}$, grey). The residual rate of Greenland MB anomaly (blue) based on having linearly regressed out the $GSI_{Exp}$ (grey) from the annual mean rate of Greenland mass balance anomaly (pink) is also shown. The linear trend values in the legend refer to the 1990–2012 period. Note that these values differ from those mentioned in the main text, because these are obtained after having removed the 1980–2018 linear trend from the mass balance time series as a first order approximation of anthropogenic forcing.

Our model experiment suggests that atmospheric circulation alone explains 56% (55%) of the observed GrIS SAT (Baffin Bay SST) warming during 1990–2012 (Fig. 3a, b). Areas located along the west, northwest, and north coasts of Greenland and in the north Labrador Sea show the largest contribution from atmospheric circulation toward explaining the observed SAT trends. These patterns are in line with the spatial structure of the high-pressure anomaly above the ice sheet that favours anomalous moisture fluxes to the northwest of the ice sheet[30] and subsidence over the west and southwest regions (Fig. 3b; Fig. 1d, f).

About half of the observed total Greenland ice loss is due to reduced SMB-driven meltwater runoff, which we have addressed in the nudging simulations. However, the other half is attributable to glacier dynamical imbalance[6]. Since the CISM-Glimmer model used in this study is developed to simulate SMB in relatively slow-flowing regions of the GrIS, rather than fast outlet glacier changes, we statistically assess how the melting rates of 260 Greenland outlet glaciers/ice caps[47] may be regulated by overlying wind changes. To do so, we utilize the nudging-run-derived GSI that explains significant interannual variability over 1980–2018 of 94 glaciers flanking the southwest (SW), central west (CW) and northwest (NW) of the island (Supplementary Fig. 7a; $r_{average} = 0.43$, with up to $r_{maximum} = 0.6$). The underlying trend in the glaciers' mass balance time series, which is a combination of the anthropogenic forcing-induced diabatic warming and the wind-driven adiabatic warming, influences the correlations and only 41 glaciers, located mostly in the SW, show significant correlations after removing

the linear trends from the data ($r_{average} = 0.39$; $r_{maximum} = 0.48$). Based on our nudging simulations, we speculate that winds may influence Greenland glacier surface melt in a similar manner as SAT and SMB. Nonetheless, targeted high-resolution model experiments are needed to support this finding.

In an effort to account for Baffin Bay upper-ocean (0–50 m) warming that influences air-sea-ice interactions, we utilize the ocean component of the nudging experiment to compare upper-ocean temperature trends in the wind-imposed simulation with the ORAS5 reanalysis during 1990–2012. We find significant upper-ocean warming in the wind-nudging simulation over the north Labrador Sea−Baffin Bay area (50°–70°W; 50°–78°N), resembling the vertical cross-section of warming along the west coast of Greenland in the reanalysis (Fig. 3c, d). A comparison of the spatially and vertically averaged upper-ocean temperature time series (Supplementary Fig. 7b) in the nudging experiment with its reanalysis counterpart reveals that about third of the trend can be simulated in our model via the sole imposition of Arctic winds (>60°N). This warming seems to be closely related to the surface net fluxes from the atmosphere into the upper-ocean ($Q_{net}$; Supplementary Fig. 7c) with a prominent lead-lag relationship developing in early summer $Q_{net}$ that leads ocean temperatures until September in both the reanalysis and the nudging experiment (Supplementary Fig. 8). These findings suggest that atmospheric circulation plays an important role in governing Baffin Bay upper-ocean temperatures. Similar findings have recently been shown for other parts of the Arctic Ocean[48]. We note that those ocean waters that primarily force Greenland termini retreat tend to be situated at the grounding lines of the glaciers[49] (>300 m), whose depths extend beyond the scope of this study. However, climate models' inability to simulate observed wind-driven circulation over the GrIS likely translates into uncertainties involving the simulation of Atlantic Water intrusion into the Baffin Bay, which is of importance, for example, to the destabilization of GrIS outlet glaciers[49–51] and thus warrants future analysis.

Lastly, motivated by the results from the nudging experiment, we estimate the total Greenland ice loss acceleration driven by the changes in the GSI and its contribution to the increasing rate of sea-level rise. The summer GSI obtained from the nudging-run accounts for about 40% of the interannual variability in the GrIS annual mean mass balance (MB)[6] whether or not the underlying trend is removed over 1980–2018 ($r_{GSI;MB} = -0.67$ (−0.62) for raw (linearly detrended) data). Similar correlations using detrended and raw data motivates the construction of a simple linear regression model to quantify the GSI-driven Greenland ice mass loss acceleration since 1990. After regressing out the summer GSI time series derived from the nudging experiment from the GrIS annual mean rate of mass balance anomaly over 1980–2018 (Fig. 3e), we find that the GSI contributed ~54% of the GrIS mass loss acceleration over 1990–2012 regardless of using linearly detrended or raw data to construct our regression model. This indicates that a substantial portion, ~71.7 Gt yr$^{-1}$ decade$^{-1}$ (out of the ~132.8 Gt yr$^{-1}$ decade$^{-1}$) total ice mass change equalling ~0.2 mm yr$^{-1}$ decade$^{-1}$ sea-level rise acceleration related to wind-induced adiabatic warming between 1990–2012, which holds potential for atmospheric circulation to affect the rate of sea-level rise to a similar extent in the coming decades.

## Fingerprints of remote forcing on the GrIS since AD 1602

We have discussed challenges in quantifying the relative contributions from internal/forced sources behind the observed GrIS ice loss and consequent sea-level rise. We follow-on this discussion first by studying the extent to which the local GrIS circulation variability is excited by remote forcing, i.e. teleconnections induced by tropical sea surface temperature (SST) anomalies. Examining the large-scale picture is important to shed more light on the sources of model biases in simulating the local GrIS circulation-surface coupling, since known

limitations of climate models to replicate tropical-Arctic linkages are likely to play a role in the aforementioned model uncertainties[15]. Then we assess whether this large-scale mechanism is stable beyond the observational era by utilizing paleoclimatic proxy-data-assimilated model experiments and ice core/oceanic coral derived temperature proxies over the past 400 years. Through these analyses, we strive to refine the pathway by which large-scale winds impact GrIS melt, particularly during the period before 1850 under less anthropogenic forcing relative to present day, as a next step to contextualize the model sensitivity issues.

In an effort to address these complexities we utilize that besides describing the local atmosphere-GrIS coupling, the GSI inherently contains information from the rotational feature of large-scale atmospheric circulation. This enables us to study how the local atmospheric circulation variability may be excited by forcing from the lower latitudes. The linearly detrended correlations between the GSI and global Z500 and SST, despite differences in the magnitudes between the JJA and annual mean correlations (Supplementary Fig. 9), show features of a teleconnection linking tropical Pacific cold SST anomalies with anomalous Greenland warming as seen in previous studies[32,34,35]. Using maximum covariance analysis (MCA, Methods, Supplementary Discussion), we reveal the primary internal coupled mode of variability between Z500 and SST. The MCA(1) spatial patterns of Z500 (Fig. 4a) and SST (Fig. 4b) exhibit a hemispheric teleconnection bridging atmospheric circulation variability over the GrIS with opposite sign (negative Pacific Decadal Oscillation (PDO)-like) changes in the tropical Pacific, and same-sign (positive Atlantic Multidecadal Oscillation (AMO)-like) changes in the tropical Atlantic. This mode explains ~55% co-variability between large-scale circulation and SST (Fig. 4c) and shows robust correspondence with the observed rate of Greenland ice loss especially between 1990–2012 (r~0.6). The regression of 200hPa streamfunction onto the Z500 expansion coefficient time series (Fig. 4a) resembles the Pacific-Arctic (PARC) teleconnection[34] and hence indicates the key role of tropical forcing in forming the mid-tropospheric high-pressure pattern over Greenland through the propagation of stationary Rossby-waves[32] (Fig. 4a). This result, alongside a regression of Z500 time series on the year-to-year variability of 200hPa horizontal winds and GrIS SAT in Fig. 4d support the idea that a substantial portion of the observed local GrIS high-pressure is likely driven by remote tropical forcing, however the quantification remains uncertain (see Supplementary Discussion).

Previous studies have raised concerns about non-stationary features of tropical-Arctic teleconnections[52] that cast doubt on the reliability of using this framework to guide future climate projections over Greenland. Therefore, we address this point by evaluating the observed teleconnection's impact on GrIS warming over centennial time scales using the Ensemble Kalman Filter 400 (EKF400) paleo-reanalysis version 2[53] spanning 1603–2002 AD. This dataset was created by assimilating early instrumental temperature, surface pressure and precipitation observations, temperature and moisture sensitive proxies from tree-ring measurements into the ECHAM5.3 atmospheric general circulation model.

Having repeated the MCA between Z500 and SST, the leading internal SST-Z500 coupled mode in the EKF400 explains nearly the same (~50%) covariance as the PARC and features strikingly similar spatial patterns to their observed counterparts (Fig. 5a, b vs. Fig. 4a, b). To further emphasize the dynamical linkage between the tropics and Greenland, we regress the 200hPa streamfunction from EKF400 onto the Z500 expansion coefficient time series. The resulting regression map closely resembles the one seen using reanalysis (Fig. 5a vs. Fig. 4a). Furthermore, the significant correlations between the SST MCA(1) time series and the PDO* (~−0.9, Methods) or the AMO* (~0.4, Methods) indices calculated from the EKF400 as well as an Atlantic Meridional Overturning Circulation (AMOC) index reconstruction[54] altogether suggest that the notable multidecadal low-frequency variabilities seen

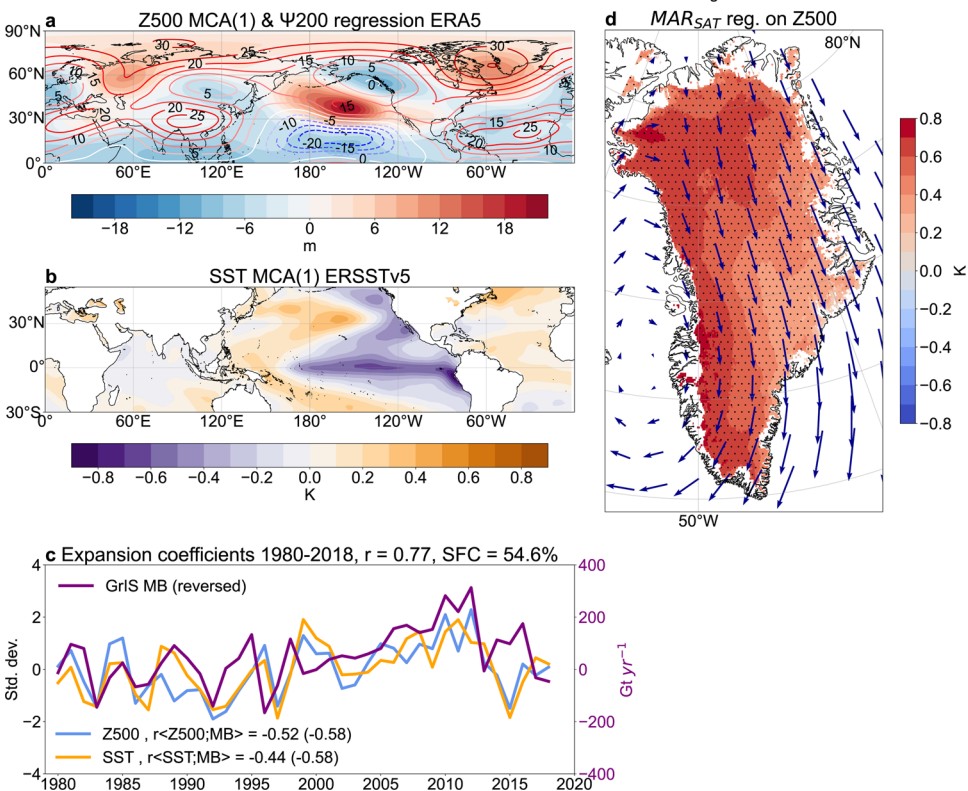

**Fig. 4 | The contribution of remote forcing to high-pressure driven adiabatic processes over the GrIS during 1980–2018.** Spatial patterns corresponding to the leading internal mode of co-variability between Northern Hemisphere annual mean **a** ERA5 500hPa geopotential height (Z500) and **b** sea surface temperatures (ERSSTv5, SST) (30°S–55°N). The corresponding expansion coefficients (EC, shown as units of standard deviations) for Z500 (blue) and SST (ERSSTv5, orange) as revealed by maximum covariance analysis (MCA, Methods) also shown in **c**. The shared fraction of covariance (SFC) is 54.6% and the correlation between the ECs is r = 0.77. The area-weighted global mean is removed before MCA from Z500 and the 60°S–60°N mean from SST. Panel **c** also shows the spatially averaged GrIS mass balance (MB) anomaly time series from The IMBIE Team[6] (purple) and the correlation between the MB and the Z500/SST expansion coefficients over 1980–2018 in the legend (with brackets showing correlations over 1990–2012). In **a**, the regression of Z500 EC(1) onto the 200 hPa streamfunction (Ψ200) calculated from ERA5 is shown with contours overlaid (unit: $10^6 m^2 s^{-1}$). Panel **d** shows the regression of Z500 EC(1) onto the MAR-simulated (Methods) surface air temperatures (SAT) (shading) and ERA5 200hPa horizontal winds (blue arrows) for the GrIS.

in the Z500 and SST MCA(1) time series are analogous to these well-known indices of multidecadal climate variability. In addition, 15 out of the 30 ice core $\Delta^{18}O$ records available across the ice sheet, and 10 out of 33 available oceanic coral $\Delta^{18}O$ records from PAGES2k show statistically significant correlations with the EKF400-derived GSI (triangles in Fig. 5c, d; Supplementary Table 1, Methods). These observational constraints along with the regression maps of the Z500 expansion coefficient time series onto the SAT and 200hPa horizontal winds over the ice sheet (Fig. 5d) closely match the circulation-surface coupling seen in the observations, and together bolster confidence that the observed large-scale wind-driven Greenland changes are part of a mode of large-scale variability that is consistent over at least the past 400 years. To verify the EKF400 results, we repeat the MCA using the Last Millennium Reanalysis[55] (Supplementary Discussion). Although discrepancies exist between the two paleo-reanalyses (Supplementary Figs. 11–12), they altogether support the idea that the significant enhancement of GrIS melting between 1990 and 2012 and associated acceleration in the rate of sea-level rise have been a manifestation of low-frequency variability in the climate system, predominantly arising from natural variability[27,34].

## Discussion

Our wind-nudging experiment highlights the contribution of large-scale winds to rapid Greenland warming, the acceleration in GrIS mass loss and related sea-level rise since the early 1990s through a tropically-excited teleconnection that modulates the high-pressure driven adiabatic warming over the ice sheet. This constitutes a new framework for tropical decadal variability to influence global sea-levels through GrIS-related barystatic sea-level rise in addition to regulating ocean thermal expansion[56,57]. The out-of-phase relationship between tropical Pacific SST and Greenland SAT anomalies associated with the PARC teleconnection results in accelerated GrIS melt and thus an increase in sea-levels during PDO-negative, which counteracts the coherent decrease in ocean thermal expansion and the increase in ocean-to-land water transport yielding a drop in sea-levels[58]. This may explain why GrIS-driven barystatic sea-level rise shows acceleration since the 1990s, whereas the rate of thermal expansion-related sea-level rise is rather constant[46]. Although the quantification of the tropical forcing in regulating the local wind-driven process remains unclear (Supplementary Discussion), evidence based on paleo-reanalyses and proxy records lends temporal credibility to the persistence of the abovementioned pathway.

To further our previous discussion on the forced/internal nature of observed GrIS surface and overlying circulation changes, we show the contrast between temperature changes over the GrIS in ERA5 and in the forced response in the CESM-LE (ensemble mean) by examining the summertime trend in temperature, geopotential height, and vertical motion (omega) zonally averaged over the ice sheet (Supplementary Fig. 10). In contrast to the vertical structure of temperature and geopotential height changes accompanied by significant downward motion in the lower troposphere in ERA5 (Supplementary Fig. 10a), the 40-member ensemble mean reflects anthropogenically-

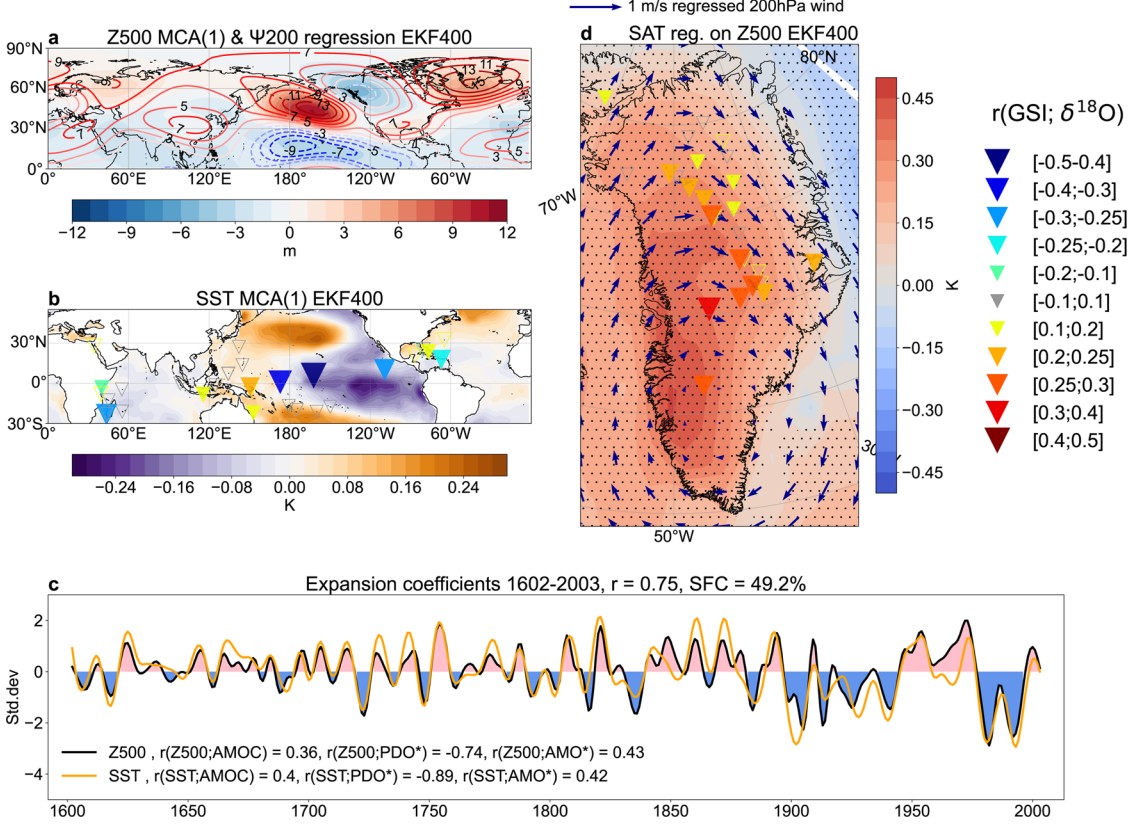

**Fig. 5 | The temporal consistency of the observed teleconnection influencing Greenland over AD 1602–2003 as simulated by the EKF400.** Spatial patterns corresponding to the leading internal mode of co-variability between Northern Hemisphere annual mean **a** 500hPa geopotential height (Z500) and **b** sea surface temperatures (SST) (30°S–55°N) and **c** the corresponding expansion coefficients (EC, shown as units of standard deviations) for Z500 (blue) and SST (orange) as revealed by maximum covariance analysis (MCA, Methods) in the Ensemble Kalman Filter (EKF400) paleo-reanalysis over 1602–2003. The ECs are filtered with a 10 yr lowpass filter to elucidate decadal variability. The shared fraction of covariance (SFC) is 49.2%, the correlation between the ECs is r = 0.75. The area-weighted global mean is removed before MCA from Z500 and the 60°S–60°N mean from SST. Panel **c** also shows the correlations between the (non-filtered) ECs and an Atlantic Meridional Overturning Circulation (AMOC) reconstruction[54] and the EKF400-derived

Pacific Decadal Oscillation (PDO) and Atlantic Multidecadal Oscillation (AMO) indices (Methods) in the legend. In **a** the regression of Z500 EC(1) onto the 200 hPa streamfunction ($\Psi$200) calculated from EKF400 is shown with contours (unit: $10^6 m^2 s^{-1}$). Panel **d** shows the regression of Z500 EC(1) onto SAT (shading) and 200 hPa horizontal winds (blue arrows) for the GrIS in the EKF400, while the triangle markers correspond to the correlation between the GSI derived from EKF400 and 30 individual ice core records across the ice sheet with filled triangles indicating significant correlations ($p < 0.05$) (also see Supplementary Table 1). Hatching indicates statistically significant regression values ($p < 0.05$). In **b**, the triangles correspond to the correlation between the 200 hPa Greenland streamfunction index (GSI) derived from EKF400 and 33 individual coral records from PAGES2k[75] (see Supplementary Table 1) with filled triangles indicating significant correlations ($p < 0.05$).

induced warming processes by simulating vertically uniform temperature and geopotential height changes associated with upward vertical motion (Supplementary Fig. 10b). Imposing ERA5 winds in the CESM1, however, captures the observed vertical temperature and geopotential height structures as well as the downward motion (Supplementary Fig. 10c) albeit with differences in the boundary layer likely related to vertical resolution of the CESM1. These features are also seen on the corresponding correlation maps between the GSI and the zonally averaged temperature over the ice sheet (Supplementary Fig. 10d–f). This analysis may offer a guide to future efforts targeting the nudging of winds in other climate models—possibly at higher resolution to better account for boundary layer processes, e.g. inversions, which are suggested to be a key contributor to GrIS surface changes[59]—, and also points to a possible source of model deficiency that needs further attention and evaluation to better constrain the forced temperature response over the ice sheet.

Improved simulation of the GrIS's observed sensitivity to wind-driven changes in climate models may thus provide a means of significantly improving predictions of the GrIS's future contribution to global environmental crises, including freshwater discharge into the Atlantic Ocean that may influence the recently observed AMOC slow-

down[54,60], global sea-level rise[1,7,61] and toxic mercury export from the ice sheet to the global oceans[62]. We hypothesize that the local atmospheric circulation bias may also be linked to model limitations in simulating tropical-Arctic teleconnections and their sensitivity to low-frequency tropical SST variability and/or high-latitude sea-ice-ocean-atmosphere interactions[15,19], which, according to previous studies[63,64], could arise from biases in simulating low-frequency SST and rainfall variability over the tropical eastern Pacific and the climatological mean flow over the North Pacific.

In a recent study Hofer et al. showed[4] that improvements in CMIP6 models, partially related to better radiative (cloud) feedbacks[22], led to simulating unprecedented GrIS melting rates by 2100 even without substantial improvements in modelling connections of large-scale circulation with local warming processes in the Arctic and over Greenland[16,18,19]. Thus, the misrepresented sensitivity of Greenland to large-scale winds in climate models prioritizes a need to refocus model evaluation efforts from expecting the models to match observed surface warming rates in their forced responses and instead assess model skill in simulating the observed sensitivity of the GrIS to overlying circulation changes. Identifying those CMIP5/6 models that have a better representation of Arctic atmospheric circulation—even if these

models are currently regarded as 'low sensitivity models'[65]—may help to provide alternative, dynamically coherent future GrIS climate projections that can enable more effective adaptation and mitigation plans to sea-level rise.

## Methods

### IMBIE 2019 Greenland ice sheet mass balance data

We use the rate of annual Greenland ice sheet mass balance data from the IMBIE 2019 Greenland Dataset as the satellite-based observational estimate of ice mass changes in the GrIS over 1980–2018. The dataset represents reconciled mass balance estimates from three independent satellite-based techniques (gravimetry, altimetry and input-output method) that are part of the Ice Sheet Mass Balance Inter-comparison Exercise (IMBIE[6]).

### Modéle Atmosphérique Régional (MAR)

We use model output from the MAR, which is a regional climate model[66] specifically designed and physically optimized for polar areas[67]. The MAR combines atmospheric modelling[66] with the Soil Ice Snow Vegetation Atmosphere Transfer Scheme[37] and has been thoroughly evaluated and used to simulate surface energy balance and mass balance processes over the GrIS[38,68]. We use model version 3.11 6-hourly forced at its lateral boundaries (temperature, specific humidity, wind speed, pressure, sea surface temperature, and sea-ice concentration) by ERA5 reanalysis at $1 \times 1 km^2$ spatial resolution at monthly time steps during 1980–2018. We use the SMB and near surface temperature variables from the MAR simulations' monthly means. The MAR near surface temperature fields are considered to be the closest match (relative to ERA-I or ERA5 reanalyses) to satellite-observed GrIS surface temperatures[39]. The SMB is calculated in the MAR according to Eq. 1.

$$SMB = precipitation_{total} - runoff_{meltwater} - sublimation/evaporation \quad (1)$$

### Reanalysis, satellite and large ensemble climate model data

We use the ERA5 reanalysis[40] and the Extended Reconstructed Sea Surface Temperature (ERSST) dataset version 5[69] from 1980 to 2018 and the Clouds and the Earth's Radiant Energy System (CERES) Energy Balanced and Filled (EBAF[44]) edition 4.1 monthly means of surface net long- and shortwave data for a satellite-based, simplified (excluding latent and sensible heat fluxes, indicated by SEB*) estimate for Greenland surface energy balance for 2001–2020. EBAF is considered to be a leading benchmarking tool for evaluating the Arctic radiative budget in model simulations[70]. We also use the ORAS5 ocean reanalysis product[71].

$$SEB^* = (LW_{net} + SW_{net})_{surface} \quad (2)$$

### CESM1 large ensemble simulations

We use the 40-member CESM1 large ensemble (CESM-LE[36]) to place observed GrIS changes in the context of simulated internal variability and the ensemble mean is used to approximate the forced response of the climate system to anthropogenic radiative forcing. The CESM-LE is forced by CMIP5 historical forcing[72] until 2005 and RCP8.5 forcing for 2006–2020 with slight perturbations in the initial conditions for each member to simulate internal variability of the climate system underlying the externally forced climate change.

### Greenland streamfunction/blocking index

To characterize wind-driven circulation aloft the GrIS while synchronously taking into consideration large-scale hemispheric variability, we compute the mean of the area-weighted 500hPa streamfunction calculated from the 500hPa horizontal wind field in ERA5 over the GrIS (80°–20°W; 60°–80°N) to get the Greenland streamfunction index (GSI). The 200 hPa GSI is calculated in a similar way. This GSI is similar

to and strongly correlated (r = 0.88) with the Greenland Blocking Index (GB2[16]). GB2 is commonly used to represent GrIS atmospheric circulation/blocking changes and is calculated by subtracting the area-weighted mean Z500 over the whole 60°–80°N hemispheric zonal band from the area-weighted mean Z500 over the GrIS (20°–80°W 60°–80°N). The GSI has the advantage that it considers rotational features of large-scale atmospheric circulation, which are the driving mechanisms behind atmospheric teleconnections, and we therefore use the GSI in our study.

### Nudging experiments with the CESM1 fully coupled model

First, we use the fully coupled CESM1, including atmosphere (CAM5), ocean (POP2), sea-ice (CICE4) and land (CLM4.5) components, to conduct nudging experiments to explore and quantify the influence of circulation changes on the GrIS. We use 6-hourly zonal and meridional winds from ERA5 to constrain the Arctic (>60°N) circulation in the CESM1 from the surface to top-of-atmosphere while keeping all external forcing agents (solar, GHG, aerosols, etc.) constant at their year 2000 levels. Note, that the response of CESM1 to nudged winds appeared to be insensitive to whether we imposed winds only above the boundary layer (>850 hPa) or from the surface of the model. During the nudging procedure, simulated winds are relaxed in each time step to corresponding ERA5 winds (interpolated from 6 h interval to model time steps (1800 s)) by adding an additional tendency term in the momentum equations, which is calculated as the difference between ERA5 winds and models' winds at each grid in each step. In our experiments, full nudging is utilized so that zonal and meridional winds are forced to vary exactly as observed in the model within the Arctic. In this way, we can perfectly 'replay' observed circulation variability in the Arctic atmosphere while other components respond to these specified wind changes. In addition, a long spin-up run is conducted before the 10-member nudging runs to ensure that the nudging simulation has no significant 'numerical shock' in the early period when reanalysis winds are suddenly added in the Arctic. To do so, a 150-year long external forcing-fixed (at year 2000 levels) nudging simulation is performed with the model perpetually nudged to winds of year 1979 in the Arctic in the same setting as what we will use in the following 10-member nudging runs. 150 years was deemed an adequate period for the model to well adjust to perpetually imposed reanalysis winds (year 1979) in the Arctic and to reach an equilibrium state. The model states on 1 January of the last 10 years of this spin-up are then used as initial conditions to reinitiate a set of new 10 members of 40-yr nudging simulations in which nudged winds in the Arctic began to vary from 1979 to 2020. We mainly focus on the ensemble mean of the 10 members in the CESM1 nudging experiments.

### Nudging experiments with the CESM1.0 Glimmer-CISM v.1.6 ice sheet model

Second, due to computational limitations and the need for forcing fields at high temporal resolution (3-hourly), we select the output fields derived from one member (no. 4) of the 10-member nudging experiments to drive the Community Ice Sheet Model (CISM) Glimmer version 1.6[73] and study the model variable "acab = accumulation and ablation rate [m/yr]; land ice surface specific mass balance" as a counterpart for MAR SMB. We select this member because the GSI trend generated in this member is the closest to the ensemble mean GSI trend, which we consider to best represent an averaged effect of circulation changes produced by all the members in the nudging experiment. The Glimmer model requires atmospheric 3-hourly (precipitation, solar radiation, temperature, pressure, humidity and winds at the surface) forcing fields that specify the state of the atmosphere and the radiative fluxes above the GrIS. All these 3-hourly forcing fields are generated by CESM1 in which the models' winds are nudged to ERA5 reanalysis in the Arctic (>60°N). Although the CISM-Glimmer's positive-degree-day (PDD) scheme relating surface air temperatures to

ice melting (for further details see Section 1.5 in https://www.cesm. ucar.edu/models/cesm1.0/cism/docs/glimmer.pdf) is insufficient to properly model future climate changes (because the empirical formulae used for present climate may change under climate change), it is adequate for use in our nudging approach where we only qualitatively investigate circulation-driven GrIS accumulation/ablation rates. Since the CISM-Glimmer's "acab" approximates the MAR surface mass balance, forcing the CISM-Glimmer with a member from the CESM1 fully coupled nudging simulations enables wind-driven adiabatic Greenland surface mass balance changes to be distinguished from diabatic processes. Caveats of the CISM-Glimmer nudging experiment include climatological biases in the simulation of surface mass balance and an overestimated melting over the period 1980–2018 compared to MAR (Supplementary Fig. 6) that is probably related to known limitations of the CISM-Glimmer such as the only available shallow ice-dynamics and the PDD scheme. Nevertheless, the qualitative comparison is still viable, as seen in Fig. 2 and surrounding discussions in the main text.

### Paleo-reanalyses and ice core data

We utilize two paleoclimatic data-assimilated climate model experiments, the Ensemble Kalman Fitting (EKF) 400 version 2[53] and the Last Millennium Reanalysis version 2.1[55]. Both products assimilate early instrumental observations along with proxy data including tree ring width (TRW), maximum latewood density (MXD), oceanic coral and other sediment proxy data, but only the LMR2.1 assimilates ice cores from the GrIS. For assimilated data details see refs. 53,55. We primarily focus on the EKF400 and its timescale (AD 1602–2003) because of slight differences in available variables from the two reanalyses; the advantage of EKF400 is that it provides horizontal winds at 200hPa to calculate streamfunction from, which is a key metric used to measure circulation-driven changes over the Greenland ice sheet. The main goal of the application of these extended reanalysis datasets is to provide evidence for the consistency of observed circulation-driven GrIS climate variability over centennial time scales.

When comparing the expansion coefficient time series derived from the MCA between large-scale circulation variability and sea surface temperatures (SST) in the EKF400 to known decadal SST variability indices, we compute the Pacific Decadal Oscillation (PDO*) index along with the Atlantic Multidecadal Oscillation (AMO*) index from the EKF400, marked with an asterisk in the main text. The PDO* is defined as the first principal component time series of the EOF of SST poleward of 20°N, while the AMO* is the spatially averaged SST in the North Atlantic basin over 0°–80°N after having removed the global mean SST (for both the PDO* and AMO*).

We further utilize 30 ice core records (with record lengths varying between 194 and 384 years) from the ISO2k database[74] that are not assimilated in the EKF400 and 33 coral records (Supplementary Table 1) from PAGES2k[75] that are assimilated in the EKF400[53] as the basis of observational constraints when evaluating the EKF400 simulated past 400 years Greenland surface temperatures and the tropical-Arctic teleconnection in Fig. 5b, d. We have selected the ice core and coral proxy records that were available with at least annual resolution.

### Statistical analysis and significance

Maximum Covariance Analysis (MCA[76]) is applied to reveal the maximum coupled variability in large-scale circulation and sea surface temperatures using singular value decomposition of the covariance matrix between Z500 and SST. The effective sample size ($N$) is used to determine the statistical significance in the study as computed by

$$N = M*(1 - r_1*r_2)*(1 + r_1*r_2)^{-1} \qquad (3)$$

where $r_1$ and $r_2$ are lag-one autocorrelation coefficients of each variable and $M$ is the original sample size. A 95% confidence level is used to determine the significance.

## Data availability

The MAR model simulations (ftp://ftp.climato.be/fettweis/MARv3.11), the ERA5 reanalysis (https://cds.climate.copernicus.eu/cdsapp# !/dataset/reanalysis-era5-pressure-levels?tab=overview), the Greenland satellite derived mass balance data (http://imbie.org/data-downloads/), the CESM-LE simulations (https://www.cesm.ucar.edu/ projects/community-projects/LENS/data-sets.html), the EKF400 version 2 simulation (https://doi.org/10.26050/WDCC/EKF400_v2.0) and the LMR2.1 data (https://atmos.washington.edu/~hakim/lmr/LMRv2/) are freely available using the access links above. The Source Data used in this study are available in Zenodo under accession code https://doi. org/10.5281/zenodo.7081743. The nudging experiments (due to their large size) are available upon request from Q.D.

## Code availability

Previous and current CESM1 versions are freely available at www.cesm. ucar.edu:/models/cesm2/. Fortran and Python codes of the analysis are available from D.T. on reasonable request.

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

## Acknowledgements

This project is jointly supported by Climate Variability & Predictability (NA18OAR4310424, Q.D.) and Modeling, Analysis, Predictions and Projections (NA19OAR4310281, Q.D.) programs as part of NOAA's Climate Program Office, NSF's Polar Programs (OPP-1744598, Q.D.) and the ÚNKP-20-3 (D.T.) New National Excellence Program of the Ministry for Innovation and Technology. The paper was prepared with the professional support of the Doctoral Student Scholarship Program of the Cooperative Doctoral Program of the Ministry of Innovation and Technology (D.T.) in addition to the 2019-2.1.11-TÉT-2020-00114 (D.T.) and FK135115 (D.T.) financed from the National Research, Development and Innovation fund. Computational resources used to perform MAR simulations have been provided by the Consortium des Équipements de Calcul Intensif (CÉCI), funded by the F.R.S.FNRS under grant 2.5020.11 and the Tier-1 supercomputer (Zenobe) of the Fédération Wallonie Bruxelles infrastructure funded by the Walloon Region under grant agreement 1117545. We acknowledge the CESM Large Ensemble Community Project and supercomputing resources provided by NSF/CISL/Yellowstone (https://doi.org/10.5065/D6RX99HX).

## Author contributions

Q.D., T.B. and D.T. equally conceived the study. D.T. analyzed the data, created the figures and wrote the first draft with input from Q.D. and T.B. Q.D. created the nudging simulations involving D.T., T.B. provided valuable feedback to D.T. on data analysis, Z.L. processed the ocean data. E.H. provided feedback on the GSI comparisons with the GBI. T.B. and E.H. contributed with insightful comments on the presentation and the structure of the manuscript. X.F. provided MARv3.11 outputs. X.F. and I.P. contributed with discussions and helpful comments on the text. All authors participated in revising the manuscript led by D.T., Q.D. and T.B.

## Funding

## Competing interests

The authors declare no competing interests.
