## [Peer Review File · Nature Communications]

Discrepancies between observations and climate models of large-scale wind-driven Greenland melt influence sea-level rise projectionsReviewers' Comments:

Reviewer #1:

Remarks to the Author:

General

The takeaways from this paper, assuming I have not misinterpreted, is that 1) about half of the observed Greenland surface warming and ice loss since 1990 is due to changes in atmospheric circulation (blocking) patterns; 2) without nudging, the CESM global climate model doesn't properly handle circulation (blocking) linkages with the tropics; 3) the observed blocking trend may or may not be forced; 4) the inability of models to capture blocking linkages is a cause for uncertainty in projected of sea level rise associated with Greenland melt. This study is certainly of value; the question is whether it is of sufficient impact to merit publication in Nature Communications. I found the paper difficult to follow in places, and what seems to be a key question (is the blocking trend natural variability?) is not really answered. Restructuring is needed to better build the arguments that are being made and the conclusions that are being drawn.

Specific

Title: Since a key part of this paper is about how CESM-LE fails to get at the large-scale wind (blocking) component, raising uncertainty as to future mass loss rates, shouldn't this be reflected in the title? That atmospheric circulation patterns affect Greenland melt is nothing new – we've known this for a long time - what seems to be new here is the implications of the model shortcomings.

Line 14: It is my understanding that the contribution to sea level rise from Greenland and all other land ice except the Antarctica are pretty much neck and neck – Greenland is not clearly leading the way. Please confirm.

Line 18-39: The wording here is confusing to me. A priori, it's not at all "puzzling" why climate models don't depict the observed increase in the frequency of blocking anomalies since about 1990. The mismatch, at face value, suggests that the trend is just an expression of natural variability. Just because the effects of blocking have well known impacts on Greenland melt is no reason to think that the models should depict a trend. Are the authors really trying to say that the models are not properly depicting the tropical forcing, such that (at least as an ensemble), they SHOULD be depicting an upward trend in blocking and that this implies uncertainty in Greenland's future contribution to sea level rise? The authors need to lay out their arguments more cleanly.

Line 53: Add a sentence briefly describing what MAR is – don't force the reader to search through the methods section.

Line 81: The same goes for CISM Glimmer.

Line 90: CESM-LE does not show the observed linear trend in winds. Is the argument that

the observed trend is natural variability or that it is forced, but that CESM-LE doesn't properly pick up the forced component? Sentence 1 of the abstract seems to argue for the latter.

Pages Line 94-98: Building on the previous comment, is the argument here is that CESM is getting the right trend in the Greenland ice Sheet SAT, but for the wrong reason? Assuming I have this right, help the reader by clearly stating this. This seems like an important point to me, and may merit mention in the abstract.

Line 104: Again, the wording here seems to imply that the CESM ensemble should be showing a trend in blocking. This is different from arguing that CESM doesn't properly capture LINKS (associated with internal variability) between tropical forcing and blocking (which is discussed later). Please clarify. Which is it?

Line 155: Do you mean increased SMB driven meltwater (not decreased)?

Lines 204: Try and set up the argument better at the start of this "fingerprints" section. Again, is the argument that CESM can't capture the internal variability component of links between tropical forcing and Greenland blocking or that there is a forced trend in blocking that CESM should be capturing but does not?

Line 235: Please briefly describe the EKF400 reanalysis. I was not aware of this effort. What is based on? How far back does it go? What have other studies concluded regarding its reliability? Don't leave the reader having to search.

Line 252: Is it that the observed large-scale wind-driven changes (the observed trend) are robust over at least the past 400 year or just that the tropical connections are robust? I think you mean the latter.

Section starting line 254, Implications: Earlier in the paper, it was highlighted that there was an upward trend in blocking frequency. That begs the question which the authors don't seem to address but really need to: Is the trend forced to some degree or is it simply internal variability?

Reviewer #2:

Remarks to the Author:

Review of "Large-scale winds contribute to sea-level rise by accelerating Greenland surface melt" by D. Topál et al.

This is a very useful and informative paper reporting substantial progress in understanding recent Greenland ice sheet (GrIS) melt. It addresses one of the most puzzling issues: that of climate models generally projecting GrIS mass loss and comparing well with satellite ice mass balance observations, but not simulating the known increase in frequency of blocking anomalies which are clearly linked to GrIS melt on an interannual basis. This conceptually

limits the reliability of global climate model projections. Here, the authors have developed a method in which a leading global model (CESM1) is nudged with reanalysis winds that have well-documented reliability at high latitudes (ERA5). To support their analysis the authors define a Greenland Streamfunction Index (GSI) calculated from the 500 hPa streamfunction, which strongly correlates with the well-known Greenland Blocking Index, but which better reflects the meteorological features related to teleconnections with lower latitudes. With these methods it is possible to investigate the model response to observed circulation changes separately from the anthropogenic forcing contribution.

The experimental results show that the CESM1 nudged to ERA5 winds does a reasonable simulation of interannual GSI variability, but otherwise the model does not simulate the observed relationship between GSI and observed surface air temperature. The same result holds for the surface energy balance. The analysis is very thorough, giving additional consideration to surface mass balance and Baffin Bay upper ocean warming that impacts air-sea-ice interaction. Overall, the GSI in the nudging experiment accounts for ~40% of the interannual GrIS annual mean mass balance variability, and ~54% of the GrIS mass loss acceleration over 1990-2012, irrespective of whether linearly detrended or raw data are used for the regression analysis. Finally, the authors use paleo-reanalysis data spanning ~400 years to elucidate lower-latitude teleconnection mechanisms influencing the GSI.

I found the manuscript straightforward to read and that the figures and supplemental material are clear and appropriate. This paper is important in providing substantial evidence of synoptic-scale wind impact on GrIS mass loss, with the nudging experiments and the paleo-reanalysis both being significant advances. This work is a continuation of earlier work by some of the authors (Ding, Hanna) on lower latitude teleconnections influencing the GrIS. Left unresolved is the specific deficiency in models such as CESM1 that renders them unable to simulate the Arctic-tropical teleconnections and hence the true GSI variability. The authors only speculate about this in the final section. This work does highlight the need to solve this problem for fully defensible GrIS mass loss and sea level rise projections using global models, particularly as the models make alarming projections even without realistic simulation of synoptic-scale winds.

Reviewer #3:

Remarks to the Author:

This manuscript is quite hard to follow and I don't quite understand what the main motivation here is. Part of this may be addressed with a stronger Introduction and clearer writing. As it stands, I think there is some unclear reasoning and confusing wording (see below). There are also several cases where the findings seem a little 'hyped-up', or are suggested to apply to other climate models (although this manuscript only considers one model).

The authors conduct an ensemble of 10 simulations with CESM1 nudged to observed winds in the Arctic and with external forcings held fixed at year-2000 levels. They then conduct a simulation with CISM, using one of these simulations as input (the one that looks closer to

the ensemble mean of the nudging simulations). I think the authors need to make a stronger case for why they do these particular simulations and how to interpret them. It would also make things easier for the reader if the simulations were referred to consistently throughout the text.

Although I spent a long time with this manuscript, I did not get very far through the results because I found them difficult to read. This might be my fault. However, the fact that the interpretation of the model experiments is not very clear might mean that this work is better suited to a longer article in a different journal (i.e. a more traditional format). That would allow the authors to place their work more clearly in the context of existing literature and provide more discussion on the interpretation and limitations of the simulations.

Main comments

There is some important information missing from the Methods regarding the simulations.

- Is the spin-up performed with year-2000 forcing?
- Are winds nudged at all levels of the atmosphere down to the surface? Does this directly set the fluxes on the top surface of the ice sheet?
- What inputs from the CESM nudging simulation are used to force CISM?

According to the Methods (L548), a primary goal of these simulations is investigate the role of natural internal variability (although this is not explicitly stated in the Introduction):

'A primary goal of these nudging experiments is to examine response of the GrIS to observed circulation changes over the past decades without interaction from anthropogenic forcing. We use winds as a proxy variable for internal circulation variability, because observed long-term wind changes in the Arctic appear to not be driven by anthropogenic forcing in current climate models.'

I am not convinced of the validity of this approach. It is possible that the current climate models do not capture observed wind trends due to model biases. Moreover, the nudging simulation includes anthropogenic effects, since it has forcing at year 2000, although it is not changing in time, and I'm having a hard time interpreting the physical meaning of it.

Sherman et al. (2020, cited in the manuscript) uses MCA applied to the CESM1 LENS to assess how internal variability has contributed to the melting of the GrIS. The authors should explain the benefits of their approach over Sherman et al. (2020).

The Introduction uses alternative wording regarding the goal of the simulations, 'to reveal the relative effects of observed (both local and remote) adiabatic and diabatic mechanisms' which is perhaps to somewhat sidestep the questions around internal variability. But I think equally here, it's not clear that one can neatly decompose these processes with these experiments. (The authors also do not really motivate why it is important to do such a decomposition).

But then at L163, the authors mention that the trend in the nudging-run is a combination of 'anthropogenic forcing-induced diabatic warming and the wind-driven adiabatic warming' (L163). Now I am even more confused, as I thought we were decomposing those two things! This might be a misunderstanding on my part.

A large part of the (short) motivation rests on "concerns about existing discrepancies between the mechanisms causing GrIS melt in the observational and model worlds." This statement seems rather strong, given the observational uncertainty. The authors also do not provide appropriate evidence for their statement that "the anthropogenically-forced response of GrIS surface conditions in CMIP5/6 climate models is mostly consistent with GrIS mass balance estimates from satellite-based observations." The studies cited (2,6-8) do not systemically assess CMIP5/6 climate models, let alone look at the anthropogenically-forced response versus internal variability. They mostly consider regional climate models or dynamically-downscaled models, rather than CMIP5/6 models (ref 2 assesses SMB in one free-running GCM). The manuscript should expand more on the differences between RCMs (as used for most previous GrIs studies) and GCMs. Using a GCM to look at GrIS changes of course has the downside of introducing larger climate biases than you would have in a RCM, but this isn't discussed in the manuscript.

Specific comments (not complete)

L4: 'with enabled ice sheet simulation' - is unclear and doesn't really reflect that this is a separate ice sheet simulation driven by one of the CESM1-nudged runs.

L10: 'heightens concern about a mismatch between observations and models of wind-driven adiabatic processes' - The authors have only looked at one model, so this statement seems rather exaggerated.

L27: "Although the extent to which anthropogenic forcing may influence the latter processes" - the latter processes being " large-scale natural atmospheric and oceanic circulation variability" - doesn't make sense. By definition natural variability is not influenced by anthropogenic forcing.

L101: 'These imply a misrepresentation of driving mechanisms, especially an underestimate of circulation-driven adiabatic processes, in the CESM-LE and likely in other CMIP5 and CMIP6 models.' - The last part about other models is speculative - remove.

L147 Internal atmospheric circulation variability may also counteract forced changes in the future, not only amplify them. I think the authors should make this clear to avoid hyping up the role of this process. Similarly at L202.

RESPONSE TO REVIEWERS

on Topál et al. ‘Large-scale winds contribute to sea-level rise by accelerating Greenland surface melt’

now changed to

Discrepancies between observations and climate models of large-scale wind-driven Greenland melt influence sea-level rise projections

We appreciate the thorough evaluation of our manuscript by the three referees, whose comments have greatly helped in refining and improving our original manuscript. We have addressed all of their comments, as we describe below in our point-by-point response. In the following, our responses are with blue letter color.

Before answering each of the specific questions by the Reviewers, we first summarize the main changes made to the text:

- (1) Revised title.
- (2) Revised Introduction in **new lines 14-82** to facilitate a better layout of our motivation and the necessity of the used wind-nudging simulation.
- (3) Revised first paragraph of the Results section in **new lines 85-91** to better contextualize the applied methods and datasets.
- (4) Having analysed 31 CMIP5 models (listed in new Supplementary Table 2) in New Supplementary Fig. 1 and added discussions in **new lines 116-121** to ensure our key motivation – that climate models struggle to replicate observed changes in circulation, while they simulate close-to-observed surface temperatures – suggest that the underlying driving mechanisms of the recent GrIS warming in observations and models may be different.
- (5) Significant revisions of the Results in **new lines 123-160** to aid a better interpretation of our modelling framework and its implications to advance the current understanding of driving mechanisms behind observed GrIS changes.
- (6) A new introductory paragraph in **new lines 263-275** to the last section of the Results explaining why we need to consider large-scale wind forcing and why an approach using paleoclimatic data helps to further understand internal vs. forced changes.
- (7) A new paragraph and a new Supplementary Fig. 10 added to the discussions in **new lines 345-363** where we now discuss possible dynamics behind the model errors showcased in the manuscript.
- (8) Revised the Method section, many details have been moved forward into the main text to make it easier to follow.
- (9) Updated References (some removed due to space limit, news are introduced in response to the Reviewers; see the manuscript file with tracked changes).

REVIEWER COMMENTS

Reviewer #1 (Remarks to the Author):

General

The takeaways from this paper, assuming I have not misinterpreted, is that 1) about half of the observed Greenland surface warming and ice loss since 1990 is due to changes in atmospheric circulation (blocking) patterns; 2) without nudging, the CESM global climate model doesn't properly handle circulation (blocking) linkages with the tropics; 3) the observed blocking trend may or may not be forced; 4) the inability of models to capture blocking linkages is a cause for uncertainty in projected of sea level rise associated with Greenland melt. This study is certainly of value; the question is whether it is of sufficient impact to merit publication in Nature Communications. I found the paper difficult to follow in places, and what seems to be a key question (is the blocking trend natural variability?) is not really answered. Restructuring is needed to better build the arguments that are being made and the conclusions that are being drawn.

The new lines refer to the main text without tracked changes.

We really appreciate the effort by Reviewer 1 and their time spent on evaluating our work. Their comments have helped clarifying our main points, improving readability. We have fully addressed the issues pointed out by Reviewer 1 and hope that this reassures them about the significance of our results.

Specific

Line 18-39: The wording here is confusing to me. A priori, it's not at all "puzzling" why climate models don't depict the observed increase in the frequency of blocking anomalies since about 1990. The mismatch, at face value, suggests that the trend is just an expression of natural variability. Just because the effects of blocking have well known impacts on Greenland melt is no reason to think that the models should depict a trend. Are the authors really trying to say that the models are not properly depicting the tropical forcing, such that (at least as an ensemble), they SHOULD be depicting an upward trend in blocking and that this implies uncertainty in Greenland's future contribution to sea level rise? The authors need to lay out their arguments more cleanly.

We have substantially revised the Introduction based on the comments from the three referees. We have stressed that models seem to simulate GrIS changes via a different mechanism than seen in the observations. We would expect to see that at least some members of a large ensemble, or members of a multi-model ensemble (individual models) reproduce the observed *connection* between atmospheric circulation and surface temperature around the ice sheet. That models simulate close-to observed GrIS surface changes even without simulating circulation changes is what puzzles us. This is something noted by earlier studies focusing on both CMIP5 and CMIP6 models (Hanna et al. 2018; Delhasse et al. 2021). We provide clarity in this regard through the revised Introduction (**new lines 14-82**), explaining in detail the motivations and the hypotheses behind our study.

To better outline our motivation, we have added a new Supplementary Figure (Supplementary Fig. 1, see also below as Response Fig. 1). This figure is aimed to complement Fig 1e in terms of showcasing GSI–SAT coupling differences in models and ERA5. The reason we did not add the new Supplementary Fig. 1 to the main text is that we primarily compare our nudging run with the CESM1-LE, because we need a model with the same physics to assure comparability.

New lines 60-69:

“Here we impose observed winds from the surface to the top-of-the-atmosphere into a fully coupled model (CESM1), while keeping anthropogenic forcing fixed, and compare our 10-member wind-nudging experiment to the CESM1 40-member large ensemble (CESM-LE³⁶). This comparison reveals the importance of observed winds in causing the recent acceleration of GrIS melt and corresponding increase in the rate of barystatic sea-level rise relative to simulated GrIS changes in the ensemble-mean (forced response) and the ensemble spread (internal variability) in the CESM-LE. The comparison between our nudging-run and the CESM-LE mean aid a better understanding of the relative importance of internal/external forcing of observed wind changes, but only to a certain extent, because we also need to consider that the nudged observed winds may already show an anthropogenic forcing signature.”

We have also included 31 CMIP5 models (listed in new Supplementary Table 2) in the analysis to show that the sensitivity issue is not only present in CESM1, but across many models as we have showed in our previous works (Ding et al. 2017; Topal et al. 2020). This issue is also present in CESM version2 as pointed out by Noël et al. (2020). Our point is that in all general circulation models (including also the CMIP6 one according to Delhasse et al. 2021) the GSI and GrIS SAT do not seem to couple as strongly as in ERA5 (Response Fig. 1). However, to rigorously quantify the uncertainty this sensitivity issue may bring to GrIS melting projections, we need to perform a model experiment with winds imposed into the CESM1. We have explained this better in the revised manuscript in the Introduction in **new lines 41-82** and also explain why it is challenging to separate internal vs external driving factors behind observed GrIS surface changes, and how this problem may be accounted for with the use of paleoclimatic data-assimilated model results.

New lines 69-82:

“However, the quantification of forced wind changes might be affected by model structural biases, which makes the precise quantification of the internal/forced observed wind changes challenging. Thus, we also compare our nudging experiment to simulations of 31 climate models participating in the Coupled Model Intercomparison Phase 5 (CMIP5), which helps to clarify whether any structural bias in the CESM-LE may be common across other models. In addition, we extend our analysis beyond the observational era by utilizing two recently available paleoclimatic proxy data-assimilated reconstructions (paleo-reanalyses) in addition to independent Greenland ice core and oceanic coral proxy records spanning the past 400 years. These analyses offer the opportunity to derive further insights into the persistence and robustness of the observed wind-driven GrIS warming mechanisms during periods with much less influence from anthropogenic emissions. By doing so we expect to advance the current understanding of large-scale atmospheric forcing driven GrIS surface changes since the early 17th century and contribute to contextualizing uncertainties of current model

projections of sea-level rise.”

Response Fig. 1. (a) Greenland ice sheet (GrIS) surface air temperatures (SAT) in ERA5, the 40 member CESM1-LE (ensemble mean with thick light blue line) and in 31 CMIP5 models (multi-model ensemble mean with thick grey line) between 1980-2018 in June-July-August (JJA). (b) the same for the Greenland Streamfunction Index calculated from 500hPa horizontal winds, and (c) scatter plot of the trends in the GSI and the GrIS SAT in each member of the CESM1-LE (seagreen triangles; ensemble mean with larger marker), each CMIP5 model (grey markers; ensemble mean with larger marker) and in ERA5 (red 'x'). Note how each CMIP5 and CESM1-LE member simulates GrIS warming without concomitant changes in their simulated GSI.

Title: Since a key part of this paper is about how CESM-LE fails to get at the large-scale wind (blocking) component, raising uncertainty as to future mass loss rated, shouldn't this be reflected I the title? That atmospheric circulation patterns affect Greenland melt is nothing new – we've known this for a long time - what seems to new here is the implications of the model shortcomings.

We thank the suggestion from the Reviewer, which we have implemented. The title has been modified accordingly to '**Discrepancies between observations and climate models of large-scale wind-driven Greenland melt influence sea-level rise projections**'. We agreed that the title now better reflects the main purpose of this research.

Line 14: It is my understanding that the contribution to sea level rise from Greenland and all

other land ice except the Antarctica are pretty much neck and neck – Greenland is not clearly leading the way. Please confirm.

Thank you for your note, we have reworded ‘dominant’ to ‘significant’ in **new line 14**.

Line 53: Add a sentence briefly describing what MAR is – don’t force the reader to search through the methods section.

Thank you for the note. We have expanded the first paragraph of the Results section and described the datasets and simulations before starting off with describing the Results in **new lines 85-91**:

“We first describe past changes in GrIS surface conditions using mass balance estimates from the Ice Sheet Mass Balance Inter-comparison Exercise (IMBIE⁶) in addition to surface mass balance (SMB, Eq. 1 in **Methods**) and surface air temperature (SAT) simulations from a widely used and GrIS optimized³⁷⁻³⁹ regional climate model, Modéle Atmosphérique Régional, (MAR, **Methods**), which is 6 hourly forced by the ERA5^{39,40} reanalysis at its boundaries. We also characterize synchronous changes in the overlying atmospheric circulation in ERA5 since 1980 alongside increasing GrIS mass loss^{3,6}.”

Line 81: The same goes for CISM Glimmer.

In **new lines 123-134**, we have pasted descriptions from the original Methods section to the main text to assure better readability, and restructured the introductory paragraph to the Results section:

“To distinguish between the two dominant mechanisms of observed GrIS summer warming as manifested in the circulation-driven adiabatic component (vertically non-uniform warming) and the radiative forcing-induced diabatic warming (vertically uniform warming), we take a two-step approach in a dynamical modelling framework: first, we use the fully-coupled Community Earth System Model 1.2 (nominal 1 degree resolution), and then we employ the Community Ice Sheet Model (CISM) Glimmer with a higher spatial resolution (~5 km) to conduct atmospheric wind-nudging experiments without interaction from time-varying anthropogenic forcing. To do so, we set external forcing (greenhouse gases, aerosols, solar) to constant values at the level of the year 2000 (367 ppm), which roughly represent the climatological mean values over 1980-2018. This allows us to directly compare the nudging experiment with ERA5, MAR and the CESM-LE to reveal the relative responses of the GrIS to atmospheric circulation changes and greenhouse gas forcing (see further details in **Methods**).”

Line 90: CESM-LE does not show the observed linear trend in winds. Is the argument that the observed trend is natural variability or that it is forced, but that CESM-LE doesn’t properly pick up the forced component? Sentence 1 of the abstract seems to argue for the latter.

Line 94-98: Building on the previous comment, is the argument here is that CESM is getting the right trend in the Greenland ice Sheet SAT, but for the wrong reason? Assuming I have this right, help the reader by clearly stating this. This seems like an important point to me, and may merit mention in the abstract.

We discuss these questions at once, since they are strongly related to changes we have made in the Introduction (**new lines 14-82**) and in the Results section (**new lines 123-160**). We have made our points clearer throughout the text and also rewrote a corresponding part in the Abstract (**new lines 2-4**): it is the sensitivity of the GrIS to circulation that we reference. We hope the question by the Reviewer has been answered above with the revised Introduction. Also, in **new lines 123-160** we have substantially changed the text according to the suggestions from the Reviewer to better lead our discussion and explain the necessity of the nudging experiments.

Line 104: Again, the wording here seems to imply that the CESM ensemble should be showing a trend in blocking. This is different from arguing that CESM doesn't properly capture LINKS (associated with internal variability) between tropical forcing and blocking (which is discussed later). Please clarify. Which is it?

To clarify, we suggested that the CESM ensemble members should capture the observed GrIS SAT–GSI coupling in order to provide us with reliable projections. During revising the main text, we kept in mind that in the original manuscript we had not set the stage properly for the reader to follow our discussion regarding the complexities behind forced vs. internal and adiabatic vs diabatic changes. In our previous replies, we hope that have answered this question also from the Reviewer.

In this paper we do not assess thoroughly the capability of the CESM1 to capture tropical-Arctic teleconnections, since other work has shown that CESM1 and other models have limitations capturing these links. Thus, here we focus on the biased sensitivity of the GrIS to overlying circulation changes in CMIP5 models (and the CESM-LE), which can be linked with the tropical teleconnection biases. To better lead the discussion, we expanded the original text in **new lines 263-275**:

“We have discussed challenges in quantifying the relative contributions from internal/forced sources behind the observed GrIS ice loss and consequent sea-level rise. We follow-on this discussion first by studying the extent to which the local GrIS circulation variability is excited by remote forcing, i.e., teleconnections induced by tropical sea surface temperature (SST) anomalies. Examining the large-scale picture is important to shed more light on the sources of model biases in simulating the local GrIS circulation-surface coupling, since known limitations of climate models to replicate tropical-Arctic linkages are likely to play a role in the aforementioned model uncertainties¹⁵. Then we assess whether this large-scale mechanism is stable beyond the observational era by utilizing paleoclimatic proxy-data assimilated model experiments and ice-core/oceanic coral derived temperature proxies over the past 400 years. Through these analyses, we strive to refine the pathway by which large-scale winds impact GrIS melt, particularly during the period before 1850 under less anthropogenic forcing relative to present day, as a next step to contextualize the model sensitivity issues.”

Lines 204: Try and set up the argument better at the start of this “fingerprints” section. Again, is the argument that CESM can't capture the internal variability component of links between tropical forcing and Greenland blocking or that there is a forced trend in blocking that CESM should be capturing but does not?

Thank you for your suggestion, we have elaborated on this in **new lines 263-275**, please also see our previous reply.

Line 155: Do you mean increased SMB driven meltwater (not decreased)?

Yes, we have made this correction.

Line 235: Please briefly describe the EKF400 reanalysis. I was not aware of this effort. What is based on? How far back does it go? What have other studies concluded regarding its reliability? Don't leave the reader having to search.

We have expanded the paragraph describing the paleo reanalysis in **new lines 300-307**:

"Therefore, we address this point by evaluating the observed teleconnection's impact on GrIS warming over centennial time scales using the Ensemble Kalman Filter 400 (EKF400) paleo-reanalysis version 2⁵³ spanning 1603-2002 AD. This dataset was created by assimilating early instrumental temperature, surface pressure and precipitation observations, temperature and moisture sensitive proxies from tree-ring measurements into the ECHAM5.3 atmospheric general circulation model."

We refer the Reviewer to Valler et al. (ref.⁵³ in the main text) who discusses the reliability of the simulations.

Line 252: Is it that the observed large-scale wind-driven changes (the observed trend) are robust over at least the past 400 year of just that the tropical connections are robust? I think you mean the latter.

In **new lines 319-329**, we add to our conclusions that between 1990-2012, the wind changes are predominantly of internal origin. However, we also keep in mind throughout the text, that one cannot give a conclusive answer without further rigorous model experiments.

"These observational constraints along with the regression maps of the Z500 expansion coefficient time series onto the SAT and 200hPa horizontal winds over the ice sheet (Fig. 5d) closely match the circulation-surface coupling seen in the observations, and together bolster confidence that the observed large-scale wind-driven Greenland changes are part of a mode of large-scale variability that is consistent over at least the past 400 years. To verify the EKF400 results, we repeat the MCA using the Last Millennium Reanalysis⁵⁵ (Supplementary Discussion). Although discrepancies exist between the two paleo-reanalyses (Supplementary Figs. 11-12), they altogether support the idea that the significant enhancement of GrIS melting between 1990-2012 and associated acceleration in the rate of sea-level rise have been a manifestation of low-frequency variability in the climate system, predominantly arising from natural variability^{27,34}."

Section starting line 254, Implications: Earlier in the paper, it was highlighted that there was an upward trend in blocking frequency. That begs the question which the authors don't seem to address but really need to: Is the trend forced to some degree or is it simply internal variability?

Regarding this very important question, we have made substantial changes in the Introduction to better set the stage for our later conclusions that question whether the observed wind-driven effects are forced or internal one cannot give a conclusive answer. We have also expanded discussions in **new lines 149-160**:

“To account for observed wind changes that may stem from anthropogenic forcing we examine wind-changes in the CESM-LE mean and in the multi-model mean of 31 CMIP5 models. Although the CESM-LE (CMIP5) mean GrIS SAT trend is ~90% (~60%) of its ERA5 counterpart, the equivalent value for the GSI is only ~10% (~2%). Compared to the significant wind changes in ERA5, the year-to-year GSI series (Fig. 1c, Supplementary Fig. 1b) show only subtle changes over 1980-2018 in both CESM-LE and CMIP5 ensemble means. These findings are similar to both the weak trend in 500hPa horizontal winds in the CESM-LE mean (Supplementary Fig. 2a) and the uniform geopotential height response simulated by other single-model large ensembles¹⁵. While <10% of observed wind changes may be affected by anthropogenic forcing, a note of caution is warranted as models may also suffer from structural biases that can obstruct a realistic wind response to anthropogenic emissions, and this shortcoming may limit the certainty of calculations that are currently available.”

We also added a new paragraph in the ‘Discussion’ section starting in **new line 331** in **new lines 345-363** and a new Supplementary Figure (Supplementary Fig. 10, see below as Response Fig. 2)

“To further our previous discussion on the forced/internal nature of observed GrIS surface and overlying circulation changes, we show the contrast between temperature changes over the GrIS in ERA5 and in the forced response in the CESM-LE (ensemble mean) by examining the summertime trend in temperature, geopotential height, and vertical motion (omega) zonally averaged over the ice sheet (Supplementary Fig. 10). In contrast to the vertical structure of temperature and geopotential height changes accompanied by significant downward motion in the lower troposphere in ERA5 (Supplementary Fig. 10a), the 40-member ensemble mean reflects anthropogenically-induced warming processes by simulating vertically uniform temperature and geopotential height changes associated with upward vertical motion (Supplementary Fig. 10b). Imposing ERA5 winds in the CESM, however, captures the observed vertical temperature and geopotential height structures as well as the downward motion (Supplementary Fig. 10c) albeit with differences in the boundary layer likely related to vertical resolution of the CESM1. These features are also seen on the corresponding correlation maps between the GSI and the zonally averaged temperature over the ice sheet (Supplementary Fig. 10d-f). This analysis may offer a guide to future efforts targeting the nudging of winds in other climate models – possibly at higher resolution to better account for boundary layer processes, e.g., inversions, which are suggested to be a key contributor to GrIS surface changes⁵⁹ –, and also points to a possible source of model deficiency that needs further attention and evaluation to better constrain the forced temperature response over the ice sheet.”

Response Fig. 2. Linear trend in zonal-mean temperature (shading), zonal-mean geopotential height (dash-dot contours; unit: m/decade) and vertical motion (omega; solid contours; unit: 10^5 Pa/s) over the GrIS (59°N–85°N;80°W–20°W) in (a) ERA5, (b) the mean of 40-members (forced component) of the CESM-LE and (c) nudging experiment between 1980–2018 JJA. Correlation between detrended zonal-mean temperature over the GrIS and 500hPa Greenland Streamfunction Index (GSI) in (d) ERA5, (e) the mean of the 40 individual correlation maps of the CESM-LE and (f) in the nudging experiment between 1980–2018 June–August (JJA).

With these added analyses and text clarifications, we aim to highlight the key findings of this study, which advance current understanding of drivers of Greenland melting and consequent barystatic sea-level rise.

References cited:

- Hanna, E., Fettweis, X., Hall, R.J.: Brief communication: Recent changes in summer Greenland blocking captured by none of the CMIP5 models. *Cryosphere* 12, 3287–3292 (2018).
- Delhasse A, Hanna E, Kittel C, Fettweis X. Brief communication: CMIP6 does not suggest any atmospheric blocking increase in summer over Greenland by 2100. *Int. J. Climatol.* 41, 2589-2596 (2021).
- Ding, Q. et al. Influence of high-latitude atmospheric circulation changes on summertime Arctic sea ice. *Nat. Clim. Chang.* 7, 289-295 (2017).
- Topál, D. et al. An internal atmospheric process determining summertime Arctic sea ice melting in the next three decades: lessons learned from five large ensembles and multiple CMIP5 climate simulations. *J. Clim.* 33(17) 7431-7454 (2020).
- Noël, B., van Kampenhout, L., van de Berg, W. J., Lenaerts, J. T. M., Wouters, B., and van den Broeke, M. R.: Brief communication: CESM2 climate forcing (1950–2014) yields realistic Greenland ice sheet surface mass balance, *The Cryosphere*, 14, 1425–1435, <https://doi.org/10.5194/tc-14-1425-2020>, 2020.

Reviewer #2 (Remarks to the Author):

Review of "Large-scale winds contribute to sea-level rise by accelerating Greenland surface melt" by D. Topál et al.

This is a very useful and informative paper reporting substantial progress in understanding recent Greenland ice sheet (GrIS) melt. It addresses one of the most puzzling issues: that of climate models generally projecting GrIS mass loss and comparing well with satellite ice mass balance observations, but not simulating the known increase in frequency of blocking anomalies which are clearly linked to GrIS melt on an interannual basis. This conceptually limits the reliability of global climate model projections. Here, the authors have developed a method in which a leading global model (CESM1) is nudged with reanalysis winds that have well-documented reliability at high latitudes (ERA5). To support their analysis the authors define a Greenland Streamfunction Index (GSI) calculated from the 500 hPa streamfunction, which strongly correlates with the well-known Greenland Blocking Index, but which better reflects the meteorological features related to teleconnections with lower latitudes. With these methods it is possible to investigate the model response to observed circulation changes separately from the anthropogenic forcing contribution.

The experimental results show that the CESM1 nudged to ERA5 winds does a reasonable simulation of interannual GSI variability, but otherwise the model does not simulate the observed relationship between GSI and observed surface air temperature. The same result holds for the surface energy balance. The analysis is very thorough, giving additional consideration to surface mass balance and Baffin Bay upper ocean warming that impacts air-sea-ice interaction. Overall, the GSI in the nudging experiment accounts for ~40% of the interannual GrIS annual mean mass balance variability, and ~54% of the GrIS mass loss acceleration over 1990-2012, irrespective of whether linearly detrended or raw data are used for the regression analysis. Finally, the authors use paleo-reanalysis data spanning ~400 years to elucidate lower-latitude teleconnection mechanisms influencing the GSI. I found the manuscript straightforward to read and that the figures and supplemental material are clear and appropriate. This paper is important in providing substantial evidence of synoptic-scale wind impact on GrIS mass loss, with the nudging experiments and the paleo-reanalysis both being significant advances. This work is a continuation of earlier work by some of the authors (Ding, Hanna) on lower latitude teleconnections influencing the GrIS. Left unresolved is the specific deficiency in models such as CESM1 that renders them unable to simulate the Arctic-tropical teleconnections and hence the true GSI variability. The authors only speculate about this in the final section. This work does highlight the need to solve this problem for fully defensible GrIS mass loss and sea level rise projections using global models, particularly as the models make alarming projections even without realistic simulation of synoptic-scale winds.

The new lines refer to the main text without tracked changes.

We thank the reviewer for their encouraging and constructive review of our work, which are complementary to comments of the other reviewers. In the revised text we attempted to more clearly address the problem of internal vs. forced changes in GrIS surface and overlying circulation, which is the main concern we extracted from the reviewer's remarks. We have changed the Introduction in **new lines 14-82** with added new thoughts on why it is challenging to separate the effects of internal versus forced changes.

We have also expanded the Results section with new thoughts addressing the concern of the Reviewer in **new lines 149-160**:

“To account for observed wind changes that may stem from anthropogenic forcing we examine wind-changes in the CESM-LE mean and in the multi-model mean of 31 CMIP5 models. Although the CESM-LE (CMIP5) mean GrIS SAT trend is ~90% (~60%) of its ERA5 counterpart, the equivalent value for the GSI is only ~10% (~2%). Compared to the significant wind changes in ERA5, the year-to-year GSI series (Fig. 1c, Supplementary Fig. 1b) show only subtle changes over 1980-2018 in both CESM-LE and CMIP5 ensemble means. These findings are similar to both the weak trend in 500hPa horizontal winds in the CESM-LE mean (Supplementary Fig. 2a) and the uniform geopotential height response simulated by other single-model large ensembles¹⁵. While <10% of observed wind changes may be affected by anthropogenic forcing, a note of caution is warranted as models may also suffer from structural biases that can obstruct a realistic wind response to anthropogenic emissions, and this shortcoming may limit the certainty of calculations that are currently available.”

We also added a new paragraph in the ‘Implications’ section starting **in new line 331 in new lines 345-363** and a new Supplementary Figure (Supplementary Fig. 10, see also above as Response Fig. 2 in our response to Reviewer#1)

“To further our previous discussion on the forced/internal nature of observed GrIS surface and overlying circulation changes, we show the contrast between temperature changes over the GrIS in ERA5 and in the forced response in the CESM-LE (ensemble mean) by examining the summertime trend in temperature, geopotential height, and vertical motion (omega) zonally averaged over the ice sheet (Supplementary Fig. 10). In contrast to the vertical structure of temperature and geopotential height changes accompanied by significant downward motion in the lower troposphere in ERA5 (Supplementary Fig. 10a), the 40-member ensemble mean reflects anthropogenically-induced warming processes by simulating vertically uniform temperature and geopotential height changes associated with upward vertical motion (Supplementary Fig. 10b). Imposing ERA5 winds in the CESM, however, captures the observed vertical temperature and geopotential height structures as well as the downward motion (Supplementary Fig. 10c) albeit with differences in the boundary layer likely related to vertical resolution of the CESM1. These features are also seen on the corresponding correlation maps between the GSI and the zonally averaged temperature over the ice sheet (Supplementary Fig. 10d-f). This analysis may offer a guide to future efforts targeting the nudging of winds in other climate models – possibly at higher resolution to better account for boundary layer processes, e.g., inversions, which are suggested to be a key contributor to GrIS surface changes⁵⁹ –, and also points to a possible source of model deficiency that needs further attention and evaluation to better constrain the forced temperature response over the ice sheet.”

We have also addressed the specific deficiency of climate models that obstructs their ability to simulate observed GSI variability by referencing our ongoing work (Feng et al. 2022, submitted) in the discussion section in **new lines 368-373**. While our work does not fully account for the underlying physical biases in climate models, we acknowledge in the paper that such diagnostics are important and should be pursued in future studies.

“We hypothesize that the local atmospheric circulation bias may also be linked to model limitations in simulating tropical-Arctic teleconnections and their sensitivity to low frequency tropical SST variability and/or high-latitude sea-ice-ocean-atmosphere interactions^{15,19}, which, according to previous studies^{63,64}, could arise from biases in simulating low-frequency SST and rainfall variability over the tropical eastern Pacific and the climatological mean flow over the North Pacific.”

Reviewer #3 (Remarks to the Author):

This manuscript is quite hard to follow and I don't quite understand what the main motivation here is. Part of this may be addressed with a stronger Introduction and clearer writing. As it stands, I think there is some unclear reasoning and confusing wording (see below). There are also several cases where the findings seem a little 'hyped-up', or are suggested to apply to other climate models (although this manuscript only considers one model).

The new lines refer to the main text without tracked changes.

We sincerely thank the Reviewer for their effort and time spent on evaluating our manuscript. **Since many comments from Reviewer#3 mirrored concerns raised by Reviewer#1, we refer Reviewer#3 to our response to Reviewer#1 starting on page 1 of this Document.**

Based on their suggestions we have substantially revised the Introduction by adding several more reasons to support our main points. We hope that the newly added parts help the Reviewer to better grasp the motivation behind and the necessity of our work. We have also included 31 other CMIP5 models to our analysis, better contextualizing previous works, (for example Hanna et al. 2018; Delhasse et al. 2021).

The authors conduct an ensemble of 10 simulations with CESM1 nudged to observed winds in the Arctic and with external forcings held fixed at year-2000 levels. They then conduct a simulation with CISM, using one of these simulations as input (the one that looks closer to the ensemble mean of the nudging simulations). I think the authors need to make a stronger case for why they do these particular simulations and how to interpret them. It would also make things easier for the reader if the simulations were referred to consistently throughout the text.

Although I spent a long time with this manuscript, I did not get very far through the results because I found them difficult to read. This might be my fault. However, the fact that the interpretation of the model experiments is not very clear might mean that this work is better suited to a longer article in a different journal (i.e. a more traditional format). That would allow the authors to place their work more clearly in the context of existing literature and provide more discussion on the interpretation and limitations of the simulations.

We thank the Reviewer for their recommended clarifications, which we have addressed in the revised text. In **new lines 14-82** we revised the Introduction by adding several additional sentences to elaborate and clarify the text. Also, we have revised the first section of the Results section in an effort to more clearly present our reasoning throughout the manuscript (**new lines 85-121**).

Main comments

There is some important information missing from the Methods regarding the simulations.
- Is the spin-up performed with year-2000 forcing?

Yes, we have added this information to the Methods section in **new lines 449-452**:

“To do so, a 150-year long external forcing-fixed (at year 2000 levels) nudging simulation is performed with the model perpetually nudged to winds of year 1979 in the Arctic in the same setting as what we will use in the following 10-member nudging runs.”

- Are winds nudged at all levels of the atmosphere down to the surface? Does this directly set the fluxes on the top surface of the ice sheet?

Yes, all model levels are nudged, which therefore sets the fluxes on the top of the ice sheet. Basically, our previous studies (Huang et al 2020, Li et al. 2022) have shown that the response of CESM1 to nudging winds is not very sensitive to how we impose wind in the low levels of the model. If we only nudge winds to some levels above the boundary layer, we still get very similar response as what we show here. We have expanded the revised Methods section in **new lines 436-440** to make this point clearer:

“We use 6-hourly zonal and meridional winds from ERA5 to constrain the Arctic (>60°N) circulation in the CESM1 from the surface to top-of-atmosphere while keeping all external forcing agents (solar, GHG, aerosols, etc.) constant at their year 2000 levels. As shown by our previous studies^{29,48}, the response of CESM1 to nudging winds is insensitive to how we impose wind in the low levels of the model.”

- What inputs from the CESM nudging simulation are used to force CISM?

The model requires atmospheric 3-hourly (precipitation, solar radiation, temperature, pressure, humidity and winds at the surface) forcing fields that specify the state of the atmosphere and the radiative fluxes above the GrIS. All these 3-hourly forcing fields are generated by the CESM1 in which the models' winds are nudged to the ERA5 reanalysis in the Arctic. We have added this information in **new lines 466-470** in the revised Method section:

“The Glimmer model requires atmospheric 3-hourly (precipitation, solar radiation, temperature, pressure, humidity and winds at the surface) forcing fields that specify the state of the atmosphere and the radiative fluxes above the GrIS. All these 3-hourly forcing fields are generated by CESM1 in which the models' winds are nudged to ERA5 reanalysis in the Arctic (>60°N).”

According to the Methods (L548), a primary goal of these simulations is investigate the role of natural internal variability (although this is not explicitly stated in the Introduction):

'A primary goal of these nudging experiments is to examine response of the GrIS to observed circulation changes over the past decades without interaction from anthropogenic forcing. We use winds as a proxy variable for internal circulation variability, because observed long-term wind changes in the Arctic appear to not be driven by anthropogenic forcing in current climate models.'

We have revised the text to avoid confusion regarding how the nudging simulations are used. Please see changes made in **new lines 41-82** in the Introduction, where we expand on the motivation behind and the necessity of the nudging simulations:

“Current understanding suggests that GrIS surface conditions are determined by a quantitatively yet uncertain combination of (i) atmospheric circulation changes primarily via the aforementioned adiabatic warming process and (ii) anthropogenic diabatic warming, which may also cause wind changes due to proportional relationships between air temperature and pressure. Nonetheless, previous analyses of state-of-the-art climate model simulations indicate weak Arctic wind changes in response to anthropogenic forcing and suggest forced temperature changes are vertically rather uniform in the summer season^{15,27}. Although one cannot rule out that model biases impact the forced temperature response in the Arctic, the observed mid-to-upper-level wind-driven process, which causes vertically non-uniform temperature changes^{15,27}, is an acknowledged contributor in shaping GrIS surface conditions during recent decades⁹⁻¹⁴.

Given the above considerations, we hypothesize that the separation of the physical mechanisms behind GrIS surface warming into adiabatic (causing vertically non-uniform warming) and diabatic (causing vertically-uniform warming) processes offers the potential for not only showcasing the divergent physical mechanisms behind observed and modelled GrIS warming, but also for quantifying the contribution of large-scale winds to accelerating sea-level rise. This is only possible by taking a conceptually different, dynamical approach contrary to previous studies that have mainly applied diagnostic and statistical approaches to observations and model simulations forced by constant or varying greenhouse gases over the past 40 to 150 years when examining the effects of atmospheric circulation on GrIS melt^{11-13,25,26}.

Here we impose observed winds from the surface to the top-of-the-atmosphere into a fully coupled model (CESM1), while keeping anthropogenic forcing fixed, and compare our 10-member wind-nudging experiment to the CESM1 40-member large ensemble (CESM-LE³⁶). This comparison reveals the importance of observed winds in causing the recent acceleration of GrIS melt and corresponding increase in the rate of barostatic sea-level rise relative to simulated GrIS changes in the ensemble-mean (forced response) and the ensemble spread (internal variability) in the CESM-LE. The comparison between our nudging-run and the CESM-LE mean aid a better understanding of the relative importance of internal/external forcing of observed wind changes, but only to a certain extent, because we also need to consider that the nudged observed winds may already show an anthropogenic forcing signature. However, the quantification of forced wind changes might be affected by model structural biases, which makes the precise quantification of the internal/forced observed wind changes challenging. Thus, we also compare our nudging experiment to simulations of 31 climate models participating in the Coupled Model Intercomparison Phase 5 (CMIP5), which helps to clarify whether any structural bias in the CESM-LE may be common across other models. In addition, we extend our analysis beyond the observational era by utilizing two recently available paleoclimatic proxy data-assimilated reconstructions (paleo-reanalyses) in addition to independent Greenland ice core and oceanic coral proxy records spanning the past 400 years. These analyses offer the opportunity to derive further insights into the persistence and robustness of the observed wind-driven GrIS warming mechanisms during periods with much less influence from anthropogenic emissions. By doing so we expect to advance the current understanding of large-scale atmospheric forcing driven GrIS surface changes since the early 17th century and contribute to contextualizing uncertainties of current model projections of sea-level rise.”

Also, in **new lines 123-160** we significantly revised the Results section where we describe the nudging simulations. This question is also referred to in our response to Reviewer#1 above.

“ To distinguish between the two dominant mechanisms of observed GrIS summer warming as manifested in the circulation-driven adiabatic component (vertically non-uniform warming) and the radiative forcing-induced diabatic warming (vertically uniform warming), we take a two-step

approach in a dynamical modelling framework: first, we use the fully-coupled Community Earth System Model 1.2 (nominal 1 degree resolution), and then we employ the Community Ice Sheet Model (CISM) Glimmer with a higher spatial resolution (~5 km) to conduct atmospheric wind-nudging experiments without interaction from time-varying anthropogenic forcing. To do so, we set external forcing (greenhouse gases, aerosols, solar) to constant values at the level of the year 2000 (367 ppm), which roughly represent the climatological mean values over 1980-2018. This allows us to directly compare the nudging experiment with ERA5, MAR and the CESM-LE to reveal the relative responses of the GrIS to atmospheric circulation changes and greenhouse gas forcing (see further details in **Methods**).

Constraining winds in the CESM1 results in the model closely resembling the interannual variability in ERA5 GSI (Fig. 1c, $r=0.81$), in MAR GrIS SAT (Fig. 1a, $r=0.84$) and in ERSSTv5 Baffin Bay SSTs (Fig. 1b, $r=0.64$). As for the summertime spatial trend patterns, on average, 53% of observed SAT, 74% of the Z500 and 35% of the Baffin Bay SST changes between 1980-2018 are captured in the nudging experiment (Fig. 1d,f). A further comparison between our nudging-run derived GrIS surface conditions and those simulated by each of the CESM-LE members reinforces the differences between the simulated and observed sensitivity of the ice sheet to wind-changes seen in other CMIP models^{4,11,42} (Supplementary Fig. 1). We highlight the contrast between the summer GrIS SAT/GSI trends over 1980-2018 in the observations and simulated in individual members comprising the CESM-LE, which underestimate the ERA5 GSI trend while encompassing the observed SAT trend during 1980-2018 (Fig. 1e). Furthermore, the correlations between GSI and GrIS spatial averaged SAT are weaker in each individual CESM-LE member than in ERA5 ($r\sim 0.75$), and only three of these members exhibit equal or greater correlations compared to the nudging experiment ($r\sim 0.6$) (Fig. 1g).

To account for observed wind changes that may stem from anthropogenic forcing we examine wind-changes in the CESM-LE mean and in the multi-model mean of 31 CMIP5 models. Although the CESM-LE (CMIP5) mean GrIS SAT trend is ~90% (~60%) of its ERA5 counterpart, the equivalent value for the GSI is only ~10% (~2%). Compared to the significant wind changes in ERA5, the year-to-year GSI series (Fig. 1c, Supplementary Fig. 1b) show only subtle changes over 1980-2018 in both CESM-LE and CMIP5 ensemble means. These findings are similar to both the weak trend in 500hPa horizontal winds in the CESM-LE mean (Supplementary Fig. 2a) and the uniform geopotential height response simulated by other single-model large ensembles¹⁵. While <10% of observed wind changes may be affected by anthropogenic forcing, a note of caution is warranted as models may also suffer from structural biases that can obstruct a realistic wind response to anthropogenic emissions, and this shortcoming may limit the certainty of calculations that are currently available.”

I am not convinced of the validity of this approach. It is possible that the current climate models do not capture observed wind trends due to model biases. Moreover, the nudging simulation includes anthropogenic effects, since it has forcing at year 2000, although it is not changing in time, and I'm having a hard time interpreting the physical meaning of it.

We have revised the Introduction and the Results sections too to better highlight our points. The main point in the Introduction that models simulate close-to observed surface temperatures over the GrIS, but only weak wind changes (both in their forced response and their ensemble members too), thus the mechanisms are likely to differ behind the observed and modeled GrIS surface warming. With having set anthropogenic forcing fixed, we do not allow the GrIS to respond to CO₂ forcing directly, thus any changes we see in the nudging simulations arise from the nudged winds, although we cannot rule out that winds themselves also carry anthropogenic forcing signals. However, based on the best modeling evidence we could get in this study and some of previous analyses (Topal et al. 2020), it appears that the contribution of global warming to long-term wind trends in the Arctic is

quite trivial. We have carefully revised the corresponding arguments in the revised manuscript as the Reviewer can see it from **new lines 14-82**, **new lines 123-160** cited above. **Also in new lines 263-275:**

“We have discussed challenges in quantifying the relative contributions from internal/forced sources behind the observed GrIS ice loss and consequent sea-level rise. We follow-on this discussion first by studying the extent to which the local GrIS circulation variability is excited by remote forcing, i.e., teleconnections induced by tropical sea surface temperature (SST) anomalies. Examining the large-scale picture is important to shed more light on the sources of model biases in simulating the local GrIS circulation-surface coupling, since known limitations of climate models to replicate tropical-Arctic linkages are likely to play a role in the aforementioned model uncertainties¹⁵. Then we assess whether this large-scale mechanism is stable beyond the observational era by utilizing paleoclimatic proxy-data assimilated model experiments and ice-core/oceanic coral derived temperature proxies over the past 400 years. Through these analyses, we strive to refine the pathway by which large-scale winds impact GrIS melt, particularly during the period before 1850 under less anthropogenic forcing relative to present day, as a next step to contextualize the model sensitivity issues.”

and in new lines 345-363:

“To further our previous discussion on the forced/internal nature of observed GrIS surface and overlying circulation changes, we show the contrast between temperature changes over the GrIS in ERA5 and in the forced response in the CESM-LE (ensemble mean) by examining the summertime trend in temperature, geopotential height, and vertical motion (omega) zonally averaged over the ice sheet (Supplementary Fig. 10). In contrast to the vertical structure of temperature and geopotential height changes accompanied by significant downward motion in the lower troposphere in ERA5 (Supplementary Fig. 10a), the 40-member ensemble mean reflects anthropogenically-induced warming processes by simulating vertically uniform temperature and geopotential height changes associated with upward vertical motion (Supplementary Fig. 10b). Imposing ERA5 winds in the CESM, however, captures the observed vertical temperature and geopotential height structures as well as the downward motion (Supplementary Fig. 10c) albeit with differences in the boundary layer likely related to vertical resolution of the CESM1. These features are also seen on the corresponding correlation maps between the GSI and the zonally averaged temperature over the ice sheet (Supplementary Fig. 10d-f). This analysis may offer a guide to future efforts targeting the nudging of winds in other climate models – possibly at higher resolution to better account for boundary layer processes, e.g., inversions, which are suggested to be a key contributor to GrIS surface changes⁵⁹ –, and also points to a possible source of model deficiency that needs further attention and evaluation to better constrain the forced temperature response over the ice sheet.”

Sherman et al. (2020, cited in the manuscript) uses MCA applied to the CESM1 LENS to assess how internal variability has contributed to the melting of the GrIS. The authors should explain the benefits of their approach over Sherman et al. (2020).

The clear benefit of our approach is that there is not any other way to account for the model bias in terms of the modeled GrIS’s sensitivity to circulation changes than a dynamical modeling framework. Sherman et al. (2020) used statistical methods to quantify internal circulation changes, but their study might be impacted by the model biases we showcase. We have revised the corresponding part in the main text in **new lines 51-59:**

“Given the above considerations, we hypothesize that the separation of the physical mechanisms behind GrIS surface warming into adiabatic (causing vertically non-uniform warming) and diabatic (causing vertically-uniform warming) processes offers the potential for not only showcasing the divergent physical mechanisms behind observed and modelled GrIS warming, but also for quantifying the contribution of large-scale winds to accelerating sea-level rise. This is only possible by taking a conceptually different, dynamical approach contrary to previous studies that have mainly applied diagnostic and statistical approaches to observations and model simulations forced by constant or varying greenhouse gases over the past 40 to 150 years when examining the effects of atmospheric circulation on GrIS melt^{11-13,25,26}.”

The Introduction uses alternative wording regarding the goal to of the simulations, 'to reveal the relative effects of observed (both local and remote) adiabatic and diabatic mechanisms' which is perhaps to somewhat sidestep the questions around internal variability. But I think equally here, it's not clear that one can neatly decompose these processes with these experiments. (The authors also do not really motivate why it is important to do such a decomposition).

It is not the internal vs. external forcing that is separable with the nudging experiment, but the adiabatic vs. diabatic changes that affect temperature. Adiabatic changes refer to large-scale motions in the atmosphere and the horizontal/vertical advection of temperature, while the diabatic term is the direct heating provided by the atmospheric response to anthropogenic greenhouse gases. We clarify such changes in the revised Introduction to better set the stage for the upcoming nudging simulations in our manuscript in **new lines 14-82**.

But then at L163, the authors mention that the trend in the nudging-run is a combination of 'anthropogenic forcing-induced diabatic warming and the wind-driven adiabatic warming' (L163). Now I am even more confused, as I thought we were decomposing those two things! This might be a misunderstanding on my part.

This sentence refers to the underlying trend in the *glaciers'* mass balance time series. We clarified this in **new lines 220-223**:

“The underlying trend in the glaciers' mass balance time series, which is a combination of the anthropogenic forcing-induced diabatic warming and the wind-driven adiabatic warming, influences the correlations and only 41 glaciers, located mostly in the SW, show significant correlations after removing the linear trends from the data ($r_{\text{average}}=0.39$; $r_{\text{maximum}}=0.48$).”

A large part of the (short) motivation rests on "concerns about existing discrepancies between the mechanisms causing GrIS melt in the observational and model worlds.' This statement seems rather strong, given the observational uncertainty.

In the revised text we changed the corresponding wording in **new lines 14-26**. We do not think it is too strong to state this, because while models simulate close-to observed surface temperatures over the GrIS (see also our newly added Supplementary Fig 1, i.e., Response Fig. 1 above) they do not simulate synchronous circulation changes. Different reanalysis products agree on atmospheric circulation changes over the ice sheet since 1980

(in MERRA2, ERA5, ERA-I etc.), as has been previously shown (see e.g. Ding et al. 2017 or Delhasse et al. 2021).

The authors also do not provide appropriate evidence for their statement that “the anthropogenically-forced response of GrIS surface conditions in CMIP5/6 climate models is mostly consistent with GrIS mass balance estimates from satellite-based observations.” The studies cited (2,6-8) do not systemically assess CMIP5/6 climate models, let alone look at the anthropogenically-forced response versus internal variability. They mostly consider regional climate models or dynamically-downscaled models, rather than CMIP5/6 models (ref 2 assesses SMB in one free-running GCM).

We have revised the wording in **new lines 16-19**:

“Indeed, the anthropogenically-forced response – multi-model means, or single-model large ensemble means – of GrIS surface conditions in CMIP5/6 climate models is mostly consistent with GrIS surface temperature and mass balance changes derived from satellite-based observations and reanalyses^{2,6-8}.”

Note, that ref.² assesses 13 climate models in the GrSMBMIP intercomparison not in one free-running GCM as the Reviewer states (<https://tc.copernicus.org/articles/14/3935/2020/tc-14-3935-2020.html>).

We have added the new Supplementary Fig 1 for the Reviewer’s information to provide evidence for that CMIP-class climate models well simulate GrIS SAT changes since 1980 (also refer to Response Fig 1 in **page 4** of the current document). In Hofer et al. 2020, they further show that CMIP6 models also well simulate observed surface changes of the GrIS even without simulating overlying circulation changes as it is shown by Delhasse et al. 2021.

The manuscript should expand more on the differences between RCMs (as used for most previous GrIS studies) and GCMs. Using a GCM to look at GrIS changes of course has the downside of introducing larger climate biases than you would have in a RCM, but this isn't discussed in the manuscript.

We thank the Reviewer for raising this point. We did use the CISM forced by our CESM1 nudging simulations to account for possible influences of the model resolution on our results. We also use MAR, which is a polar RCM (6 hourly forced by ERA5). We have clarified these in **new lines 85-91**:

“We first describe past changes in GrIS surface conditions using mass balance estimates from the Ice Sheet Mass Balance Inter-comparison Exercise (IMBIE⁶) in addition to surface mass balance (SMB, Eq. 1 in **Methods**) and surface air temperature (SAT) simulations from a widely used and GrIS optimized³⁷⁻³⁹ regional climate model, Modéle Atmosphérique Régional, (MAR, **Methods**), which is 6 hourly forced by the ERA5^{39,40} reanalysis at its boundaries. We also characterize synchronous changes in the overlying atmospheric circulation in ERA5 since 1980 alongside increasing GrIS mass loss^{3,6}”

Nonetheless, following this suggestion, we have expanded the discussion section (starting in **new lines 331**) and added a new Supplementary Figure (Supplementary Fig. 10, see also above as Response Fig. 2 in our response to Reviewer#1) in **new lines 345-363** to mention that to capture GrIS boundary layer processes new simulations may be needed at higher spatial resolutions:

“To further our previous discussion on the forced/internal nature of observed GrIS surface and overlying circulation changes, we show the contrast between temperature changes over the GrIS in ERA5 and in the forced response in the CESM-LE (ensemble mean) by examining the summertime trend in temperature, geopotential height, and vertical motion (omega) zonally averaged over the ice sheet (Supplementary Fig. 10). In contrast to the vertical structure of temperature and geopotential height changes accompanied by significant downward motion in the lower troposphere in ERA5 (Supplementary Fig. 10a), the 40-member ensemble mean reflects anthropogenically-induced warming processes by simulating vertically uniform temperature and geopotential height changes associated with upward vertical motion (Supplementary Fig. 10b). Imposing ERA5 winds in the CESM, however, captures the observed vertical temperature and geopotential height structures as well as the downward motion (Supplementary Fig. 10c) albeit with differences in the boundary layer likely related to vertical resolution of the CESM1. These features are also seen on the corresponding correlation maps between the GSI and the zonally averaged temperature over the ice sheet (Supplementary Fig. 10d-f). This analysis may offer a guide to future efforts targeting the nudging of winds in other climate models – possibly at higher resolution to better account for boundary layer processes, e.g., inversions, which are suggested to be a key contributor to GrIS surface changes⁵⁹ –, and also points to a possible source of model deficiency that needs further attention and evaluation to better constrain the forced temperature response over the ice sheet.”

Specific comments (not complete)

L4: 'with enabled ice sheet simulation' - is unclear and doesn't really reflect that this is a separate ice sheet simulation driven by one of the CESM1-nudged runs.

Thank you for the note, we have removed this from the Abstract and rewrote the corresponding lines in the Results in **new lines 123-134**:

“To distinguish between the two dominant mechanisms of observed GrIS summer warming as manifested in the circulation-driven adiabatic component (vertically non-uniform warming) and the radiative forcing-induced diabatic warming (vertically uniform warming), we take a two-step approach in a dynamical modelling framework: first, we use the fully-coupled Community Earth System Model 1.2 (nominal 1 degree resolution), and then we employ the Community Ice Sheet Model (CISM) Glimmer with a higher spatial resolution (~5 km) to conduct atmospheric wind-nudging experiments without interaction from time-varying anthropogenic forcing. To do so, we set external forcing (greenhouse gases, aerosols, solar) to constant values at the level of the year 2000 (367 ppm), which roughly represent the climatological mean values over 1980-2018. This allows us to directly compare the nudging experiment with ERA5, MAR and the CESM-LE to reveal the relative responses of the GrIS to atmospheric circulation changes and greenhouse gas forcing (see further details in **Methods**).”

L10: 'heightens concern about a mismatch between observations and models of wind-driven adiabatic processes' - The authors have only looked at one model, so this statement seems rather exaggerated.

We have also cited many studies which agreed that this is the case across many models (including many of our own works). Nevertheless, we now include 31 CMIP5 models to back up our arguments across the manuscript. We also refer to Response Fig. 1 included above (in the response to Reviewer #1)

L27: "Although the extent to which anthropogenic forcing may influence the latter processes" - the latter processes being " large-scale natural atmospheric and oceanic circulation variability" - doesn't make sense. By definition natural variability is not influenced by anthropogenic forcing.

Thank you for your comment, we have removed 'natural' from the text in **new line 29** as we meant that anthropogenic forcing may influence large-scale winds (which may be the case through the effect of diabatic forcing on temperature and thus on geopotential heights).

L101: 'These imply a misrepresentation of driving mechanisms, especially an underestimate of circulation-driven adiabatic processes, in the CESM-LE and likely in other CMIP5 and CMIP6 models.' - The last part about other models is speculative - remove.

We have revised the corresponding parts and now expanded the analysis using 31 CMIP5 models in **new lines 116-121**:

"Contrary to the observed synchrony between GrIS surface and overlying atmospheric circulation changes, 31 CMIP5 climate models and the 40-member CESM1-LE also indicate less of an influence from wind-changes on GrIS warming (Supplementary Fig. 1). It is also the case in version 2 of CESM⁴² and more generally in the CMIP6 models^{4,11}. These findings suggest the possibility that climate models may misrepresent driving mechanisms; hence, to further interpret the potential consequences, we conduct a wind-nudging model experiment (see Methods)."

L147 Internal atmospheric circulation variability may also counteract forced changes in the future, not only amplify them. I think the authors should make this clear to avoid hyping up the role of this process. Similarly at L202.

We have revised the text following the suggestion from the Reviewer in **new line 204**. In addition, we have implemented changes to the text to better highlight that our primary focus is addressing the mechanism responsible for accelerating GrIS melting seen between 1990-2012, and, moreover, to quantify the extent to which large-scale winds may have contributed to it. Given accelerated GrIS melt, this is a justifiable window to revisit the model evaluation frameworks that mainly consider surface temperature when deciding upon an accurate or inaccurate model projection.

Reviewers' Comments:

Reviewer #1:

Remarks to the Author:

This is my second review of this paper. I admire the efforts of the author to revise the manuscript in response to my comments and suggestion and those of the other reviewers. While I leave it to the Reviewer 3 to assess details of the experimental setup (this reviewer is clearly more attuned to such issues than I am), I continue to struggle with the overall framing of this paper, and as such my comments here focus on the abstract and the introduction section.

From their response to the first comment in my review, the authors seem to set the stage: "That models simulate close-to-observed GrIS surface changes even without simulating circulation changes is what puzzles us". However, in the I don't really see this coming out clearly in the revised introduction. Building on this, are the authors arguing there SHOULD be forced changes in atmospheric circulation patterns in a warming climate linked to tropical forcing that will accelerate Greenland melt, but that the models don't depict this? If so, this also still isn't coming across clearly.

The abstract doesn't help in the framing. How about something like this as the first sentence: "While coupled climate models project that Greenland melt will continue to accelerate as the climate warms, models fail to capture observed connections between Greenland melt and changes in atmospheric blocking. By imposing observed...." Then, later in the abstract, how about: "We further reveal fingerprints of this circulation connection in paleo-reanalyses spanning the past 400 years. The inability of models to properly capture wind-driven processes on Greenland melt has potential implications for projected sea level rise".

Despite the authors response to my first specific comment and related ones that follow, I'm still struggling with some of the logic flow in the introduction. Building on my previous comment, is the key argument basically that the models are getting the right answer (in terms of models simulating surface changes close to observations) but for the wrong reasons? If so, simply state this to help set the stage. How about "...mass balance changes derived from satellite-based observations and reanalyses. However, it seems that the models are getting the right answer but for the wrong reason. Previous studies suggest ..."

Also to better clarify in the introduction - is the observed trend in blocking patterns been clearly linked to a forcing? It seems to me that unless it has, there is no reason that models should depict changes - it's just internal variability. The statement on page 2, line 31, that the effects of anthropogenic forcing on atmospheric (and oceanic) circulation are unclear adds to my confusion. Note that the literature is replete with studies of trends in atmospheric circulation patterns and possible links to climate change that turn out to be ephemeral (see the literature on the NAO, for example).

Reviewer #2:

Remarks to the Author:

My overall opinion about this manuscript remains unchanged, that the methods are fundamentally sound and that the topic is appropriate for Nature Communications. As I am familiar with the authors' earlier work, and with the GIS mass loss problem in general, I was clearly less demanding of the manuscript than the other two reviewers. The other reviewers raised many valid concerns in the context of a more general readership, and the authors have thoughtfully and thoroughly addressed these concerns. I recommend acceptance of the revised manuscript for publication.

RESPONSE TO REVIEWERS

on Topál et al.

‘Discrepancies between observations and climate models of large-scale wind-driven Greenland melt influence sea-level rise projections’

We appreciate the thorough evaluation of our manuscript by the two referees, whose comments have helped in further clarifying our main argument. We have addressed all of their comments, as we describe below in our point-by-point response. In the following, our responses are with blue letter color.

REVIEWER COMMENTS

Reviewer #1 (Remarks to the Author):

This is my second review of this paper. I admire the efforts of the author to revise the manuscript in response to my comments and suggestion and those of the other reviewers. While I leave it to the Reviewer 3 to assess details of the experimental setup (this reviewer is clearly more attuned to such issues than I am), I continue to struggle with the overall framing of this paper, and as such my comments here focus on the abstract and the introduction section.

We sincerely thank the Reviewer for their second revision of our paper and for pointing out some issues left in setting the stage for our arguments. We have revised the Abstract and accepted the suggested changes in the Introduction.

From their response to the first comment in my review, the authors seem to set the stage: “That models simulate close-to-observed GrIS surface changes even without simulating circulation changes is what puzzles us”. However, in the I don’t really see this coming out clearly in the revised introduction. Building on this, are the authors arguing there SHOULD be forced changes in atmospheric circulation patterns in a warming climate linked to tropical forcing that will accelerate Greenland melt, but that the models don’t depict this? If so, this also still isn’t coming across clearly.

We accept the Reviewer’s following comments, which we have incorporated into the revised Abstract and Introduction. We removed one sentence from line 56 (in the tracked changes version) to simplify our argument and this way we hope that **new lines 42-51** is more easily to follow where we describe the problem of forced versus internal nature of wind changes over the GrIS.

Regarding whether we should or should not see anthropogenically forced changes in circulation over the GrIS, we already stated in the manuscript that (in **new lines 44-45**) ‘anthropogenic diabatic warming (...)’ ‘(...) may also cause wind changes due to proportional relationships between air temperature and pressure (...)’, hence it is physically plausible to think that there should be changes in circulation in response to anthropogenic forcing, however, we do not see such response in the models (see Refs 15 and 27, our previous works). Hence, a forced response is imaginable, but current models do not simulate

it. So, at this point we think that whether one *expects* or *does not expect* to see a circulation response to anthropogenic forcing is irrelevant.

The Reviewer's question is also about the tropical teleconnection part, our answer is no, we do not argue that there should be a forced response in atmospheric circulation that is linked with tropical forcing. We only mention the possibility of a locally generated response, via the thermodynamic energy equation. Whether tropical teleconnections reaching the GrIS interacts with anthropogenic forcing is not the topic of this paper, it would require another extensive amount of works considering existing tremendous uncertainties in that topic too (e.g., see ref 15 in the main text).

Considering suggestions from the Reviewer, we have modified the Introduction in **new lines 67-71** to clarify our arguments. We believe the reader can more clearly follow why we cannot give a conclusive answer on internal/forced nature of the wind changes over the GrIS.

The abstract doesn't help in the framing. How about something like this as the first sentence: "While coupled climate models project that Greenland melt will continue to accelerate as the climate warms, models fail to capture observed connections between Greenland melt and changes in atmospheric blocking. By imposing observed...." Then, later in the abstract, how about: "We further reveal fingerprints of this circulation connection in paleo-reanalyses spanning the past 400 years. The inability of models to properly capture wind-driven processes on Greenland melt has potential implications for projected sea level rise".

We thank the Reviewer for their excellent suggestions on the Abstract, we have significantly revised it based on their comments in **new lines 2-13**. We hope that now the message comes across more clearly.

Despite the authors response to my first specific comment and related ones that follow, I'm still struggling with some of the logic flow in the introduction. Building on my previous comment, is the key argument basically that the models are getting the right answer (in terms of models simulating surface changes close to observations) but for the wrong reasons? If so, simply state this to help set the stage. How about "...mass balance changes derived from satellite-based observations and reanalyses. However, it seems that the models are getting the right answer but for the wrong reason. Previous studies suggest ..."

Similar to the Abstract, we acknowledge the Reviewer's suggestion in **new lines 20-22**.

Also to better clarify in the introduction - is the observed trend in blocking patterns been clearly linked to a forcing? It seems to me that unless it has, there is no reason that models should depict changes – it's just internal variability. The statement on page 2, line 31, that the effects of anthropogenic forcing on atmospheric (and oceanic) circulation are unclear adds to my confusion. Note that the literature is replete with studies of trends in atmospheric circulation patterns and possible links to climate change that turn out to be ephemeral (see the literature on the NAO, for example).

We agree with the Reviewer that the statements related to the anthropogenic forcing versus internal variability may have been confusing. Thus, to simplify the Introduction, we remove that sentence since a similar point is made few rows below. Now, from **new lines 42-51** we hope it is easier to follow why it is still challenging to be conclusive on the internally versus externally forced nature of the GrIS circulation changes.

Reviewer #2 (Remarks to the Author):

My overall opinion about this manuscript remains unchanged, that the methods are fundamentally sound and that the topic is appropriate for Nature Communications. As I am familiar with the authors' earlier work, and with the GIS mass loss problem in general, I was clearly less demanding of the manuscript than the other two reviewers. The other reviewers raised many valid concerns in the context of a more general readership, and the authors have thoughtfully and thoroughly addressed these concerns. I recommend acceptance of the revised manuscript for publication.

We sincerely thank the Reviewer for their second revision of our paper and for recommending it for publication.